# Decoding Natural Images from EEG for Object Recognition

**Yonghao Song**[1], **Bingchuan Liu**[1], **Xiang Li**[1], **Nanlin Shi**[1], **Yijun Wang**[2], **Xiaorong Gao**[1*]
[1] Department of Biomedical Engineering, Tsinghua University
[2] Institute of Semiconductors, CAS
`gxr-dea@tsinghua.edu.cn`

## Abstract

Electroencephalography (EEG) signals, known for convenient non-invasive acquisition but low signal-to-noise ratio, have recently gained substantial attention due to the potential to decode natural images. This paper presents a self-supervised framework to demonstrate the feasibility of learning image representations from EEG signals, particularly for object recognition. The framework utilizes image and EEG encoders to extract features from paired image stimuli and EEG responses. Contrastive learning aligns these two modalities by constraining their similarity. Our approach achieves state-of-the-art results on a comprehensive EEG-image dataset, with a top-1 accuracy of 15.6% and a top-5 accuracy of 42.8% in 200-way zero-shot tasks. Moreover, we perform extensive experiments to explore the biological plausibility by resolving the temporal, spatial, spectral, and semantic aspects of EEG signals. Besides, we introduce attention modules to capture spatial correlations, providing implicit evidence of the brain activity perceived from EEG data. These findings yield valuable insights for neural decoding and brain-computer interfaces in real-world scenarios. Code available at https://github.com/eeyhsong/NICE-EEG.

## 1 Introduction

Our daily life relies on accurately and rapidly identifying objects in complex visual environments (DiCarlo & Cox, 2007). Researchers have pursued decoding natural images from brain activity, aiming to deepen our understanding of the brain and create user-friendly brain-computer interfaces (BCIs) (Kamitani & Tong, 2005; Kay et al., 2008; Gao et al., 2021). Functional magnetic resonance imaging (fMRI), which records blood oxygen level-dependent signals, has been a popular choice for categorizing objects observed by humans (Du et al., 2023; Horikawa & Kamitani, 2017; Allen et al., 2022). Despite its high spatial resolution, fMRI typically requires several seconds to provide a stable response to a single stimulus, limiting its real-time applicability in daily interactions (Lin et al., 2022). Similarly, Magnetoencephalogram (MEG), with high time resolution, has been employed for this purpose, but it is hindered by cost and large devices (Cichy et al., 2014; Hebart et al., 2023).

Electroencephalogram (EEG) has emerged as a valuable tool for decoding images based on visual-evoked brain activities (Spampinato et al., 2017). EEG has high time resolution, low cost, and good portability, but the low signal-to-noise ratio takes problems (Pan et al., 2022; Kobler et al., 2022). Some efforts have yielded impressive classification results and captured saliency maps using EEG signals (Palazzo et al., 2021). However, its flawed block-design experiments group images of the same class within one block, resulting in classification that relies on block-level temporal correlation rather than stimulus-related activity (Li et al., 2021). Some studies have achieved above-chance performance but were restricted to much data with one subject data (Ahmed et al., 2021). Moreover, available datasets often feature a limited number of image categories, posing challenges for real-world image decoding tasks. The latest study has significantly expanded the situation by amassing a large and diverse EEG dataset comprising 16,740 image stimuli spanning 1,854 concepts, employing rapid serial visual presentation (Gifford et al., 2022). While demonstrating the feasibility of image-to-EEG encoding and category separability, this work primarily focused on analyzing electrodes in the occipital and parietal regions, overlooking the inferior temporal cortex plays a necessary role in object recognition (Dapello et al., 2023). In addition, the work focuses on the peak of pairwise decoding about 110 ms after stimuli onset. Generally, this latency involves visual conduction and primary

visual processing by a large margin, insufficient for semantic understanding Xu et al. (2023). In short, the issues of multi-class object recognition and biological plausibility warrant further exploration.

Pattern recognition plays a pivotal role in EEG analysis. Existing work has predominantly relied on supervised learning, often dealing with limited data from a few categories (Cheng et al., 2022). Applying machine learning methods directly, such as support vector machines and deep neural networks, has proven challenging in obtaining intrinsic representations (Fu et al., 2022; Liu et al., 2021). People have performed self-supervised learning and leveraged information from other modalities in computer vision research (Wang et al., 2022). Text replaces the traditional hard labels to enable self-supervised learning to produce superior image representations, often leading to impressive zero-shot results on downstream tasks (Radford et al., 2021). Researchers have also used contrastive learning to obtain consistent embedding of neural recordings across multiple animals (Schneider et al., 2023). Building upon these insights, we explore the intriguing possibility of recognizing visual objects from EEG signals in a self-supervised manner. Another issue pertains to the limitations of commonly employed EEG feature extractors, typically structured around convolutional layers applied separately along temporal and spatial dimensions of raw EEG signals (Schirrmeister et al., 2017; Lawhern et al., 2018; Song et al., 2023). This organization disrupts the correlations between electrode channels, hindering our perceiving spatial properties of brain activity (Ding et al., 2023).

To address the above limitations, we introduce a self-supervised framework to decode image representations from EEG signals, focusing on object recognition. Our approach uses contrastive learning as the bridge connecting image stimuli and EEG responses. We feed image-EEG pairs to the model and process them by an image encoder and an EEG encoder separately. These modalities are aligned by constraining their cosine similarity, resulting in the ability to perform zero-shot EEG decoding after training, achieving remarkable classification accuracy for previously unseen image categories. With the framework, we provide an overall resolving of the visual object recognition processing in our brain from spatial, temporal, spectral, and semantic aspects with comparative experiments. The results are consistent with established neuroscientific knowledge, further demonstrating the feasibility of EEG-based image decoding. Furthermore, we present two spatial modules with self-attention and graph attention that integrate with the EEG encoder (Dosovitskiy et al., 2021; Veličković et al., 2018), helping us to emphasize key brain regions relevant to object recognition.

Our main contributions can be summarized as follows:

- We propose a self-supervised framework for EEG-based object recognition with contrastive learning. Remarkable zero-shot performance has been achieved on large and rich datasets.

- We demonstrate the feasibility of investigating natural image information from EEG signals. Extensive experiments affirm the biological plausibility, which brings a resolving of human object recognition from temporal, spatial, spectral, and semantic aspects.

- We apply two plug-and-play modules to capture spatial correlations among EEG channels, offering evidence that the model discerns the spatial dynamics of object recognition.

## 2 RELATED WORKS

Decoding visual information from our brain has been a long-standing pursuit in neuroscience and computer science (Miyawaki et al., 2008; Jia et al., 2021). While some progress has been made in decoding steady-state visual stimuli, the accurate and rapid decoding of semantic information in natural images has remained a challenge (Shi et al., 2023). fMRI has been widely used to estimate semantic and shape information from visual processing in the brain (Ho et al., 2023; Takagi & Nishimoto, 2023). However, the demand for high-speed and practical applications in brain-computer interaction necessitates alternative approaches. EEG has emerged as a promising option due to its high temporal resolution and portability (Willett et al., 2021). But general performance on different subjects and biological plausibility remain unsolved (Ahmed et al., 2021). Additionally, prior approaches have often relied on supervised learning methods with limited image categories, overlooking the intrinsic relationship between image stimuli and brain responses (Shen et al., 2019; Li et al., 2021; Liu et al., 2023). These limitations hinder their practicality in real-world scenarios with generalization to new categories. Motivated by these challenges, we present an EEG-based image decoding framework that employs self-supervised learning, enabling the model to achieve zero-shot generalization in object recognition tasks, further demonstrating the feasibility.

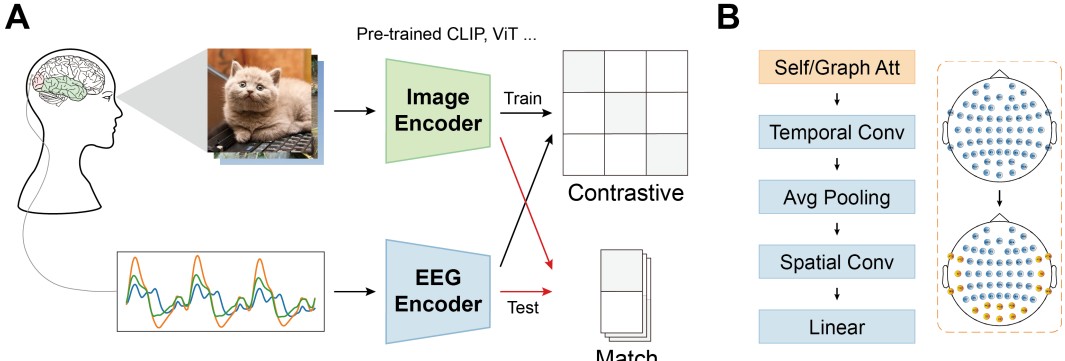

Figure 1: (A) Overall framework for EEG-based object recognition. During training, image-EEG pairs are processed by an image encoder (pre-trained) and an EEG encoder. The objective is to increase the similarity between matched pairs while decreasing it for unmatched pairs. During testing, a few unseen images of target concepts (classes) are processed in advance into templates. Then, the model obtains results by matching test data to templates. (B) Architecture of the EEG encoder. Temporal-spatial convolution is used with spatial modules, made with self and graph attention, to reveal spatial features of brain activity. The linear layer is used to project the feature dimension.

## 3 METHODS

### 3.1 OVERVIEW

We propose a self-supervised framework, Natural Image Contrast EEG (NICE), to decode images from EEG signals. The overall framework is depicted in Fig. 1(A).

During training, stimulus-response pairs comprising images and EEG signals are fed into the framework. The image and EEG encoder extract features from their respective modalities. Contrastive learning is employed to align these features, optimizing the similarity between matched pairs and reducing it for unmatched pairs. This self-supervised approach enables the model to learn shared information between the two modalities instead of traditional supervised learning with predefined labels. Before the test process, we use a few images that belong to the image concepts (classes) to be classified, and have not appeared in the EEG collection experiments. These images are processed and averaged to obtain one template for each concept. Next, the EEG encoder processes the test signals, and their similarity with all templates is calculated for matching results.

For the EEG encoder, we adapt temporal-spatial convolution (TSConv), which is widely used in EEG analysis, as shown in Fig. 1(B). In addition, we integrate two plug-and-play spatial modules with self-attention and graph attention. These modules are instrumental in preserving the spatial characteristics of EEG channels, reflecting intrinsic patterns of brain activity. Note that the image encoder has been pre-trained on other image datasets. The leading studies in computer vision help us with larger sample space, when we cannot easily collect numbers of brain responses.

### 3.2 NETWORK ARCHITECTURE

#### 3.2.1 EEG ENCODER

Researchers usually arrange the raw EEG trials into two dimensions $C \times T$, in which $C$ denotes electrode channels, and $T$ denotes time samples. Convolution along the two dimensions is widely used in deep learning-based EEG analysis models. In the same way, we present a very concise network architecture as shown in Table 1. One-dimension convolution is first applied to capture the temporal features with $k$ kernels of size $(1, m_1)$ and stride of $(1, 1)$. An average pooling layer with a kernel size of $(1, m_2)$ and stride of $(1, s)$ is introduced to alleviate overfitting and smooth the temporal features. Another one-dimension convolution is then used for spatial features keeping $k$ kernels of size $(ch, 1)$ and stride of $(1, 1)$, where $ch$ usually equals the number of electrodes. After

Table 1: Architecture of EEG encoder with SA or GA followed by TSConv

| Layer | In | Out | Kernel | Stride | Dimension |
|---|---|---|---|---|---|
| SA & GA | | | | $[(b, 1, C, T) \to (b, 1, C, T)]$, $b$ is batch | |
| Temporal Conv | 1 | $k$ | $(1, m_1)$ | $(1, 1)$ | $(b, k, C, T - m_1 + 1)$ |
| Avg Pooling | $k$ | $k$ | $(1, m_2)$ | $(1, s)$ | $(b, k, C, (T - m_1 - m_2 + 1)/s + 1)$ |
| Spatial Conv | $k$ | $k$ | $(C, 1)$ | $(1, 1)$ | $(b, k, 1, (T - m_1 - m_2 + 1)/s + 1)$ |
| Flatten&Linear | | | | $[k * ((T - m_1 - m_2 + 1)/s + 1) \to$ shape of image feature$]$ | |

each convolutional layer, batch normalization and exponential linear units (ELU) activation functions are used for better training and nonlinearity (Clevert et al., 2016). Finally, a linear layer is added as a projector to transform the features to the same size as the output of the image encoder.

### 3.2.2 IMAGE ENCODER

The image features are extracted with the pairing EEG features for contrastive learning. There have been some excellent computer vision models that can obtain image features with semantic discrimination. We prefer directly using models pre-trained on other image datasets to get a large sample space. The feature extraction parts of Contrastive Language-Image Pre-training (CLIP) (Radford et al., 2021), Vision Transfomer (ViT) (Dosovitskiy et al., 2021), and Residual Neural Networks (ResNet) (He et al., 2016) are tried separately in our framework for demonstration. All the images for stimuli can be processed in advance, thus speeding up our model training.

### 3.2.3 CONTRASTIVE LEARNING

The framework is constructed with contrastive learning, shown in Algorithm 1. The outputs of the image and EEG encoder are normalized separately. Then, the dot product is used to evaluate the similarity of all image-EEG feature pairs. A scaled temperature parameter is used to adjust distribution probability. InfoNCE loss (Radford et al., 2021; van den Oord et al., 2019) is employed as the objective function to increase the similarities between matched pairs and decrease those between unmatched pairs. After adequate training, the EEG encoder can extract representations similar to corresponding images. The self-supervised strategy allows us to learn inherent patterns from EEG signals without labels, rather than directly separate different classes with supervised learning.

---

**Algorithm 1** Natural Image Contrast EEG framework

---

1: **Input**: (Image, EEG) - stimulus & response
2: **Model**: Enc_img - CLIP & Enc_eeg - TSConv

3: # E - $(batch, channel, electrode, sample)$       ▷ batch of input EEG
4: # I - $(batch, channel, height, width)$       ▷ batch of input images
5: # $\tau$ - learned temperature parameter

6: # extract normalized representations from the raw image and EEG
7: E_f = Norm(Linear(Enc_eeg(E)))
8: I_f = Norm(Enc_img(I))       ▷ can be obtained before training

9: # scaled pairwise cosine similarity
10: logits = dot(E_f, I_f.t) * exp($\tau$)

11: # symmetric loss function
12: labels = arange(batch)
13: loss_e = cross_entropy_loss(logits, labels, axis=0)
14: loss_i = cross_entropy_loss(logits, labels, axis=1)
15: loss = (loss_e + loss_i) / 2

---

### 3.3 PLUG-AND-PLAY MODULE

#### 3.3.1 SELF-ATTENTION

We utilize two approaches, self-attention (SA) Vaswani et al. (2017), and graph attention (GA) Veličković et al. (2018), to supplement the EEG encoder as "spatial filters" to encapsulate electrode correlations and reflect the spatial dynamics of brain activity. We use SA on the electrode channels to evaluate the spatial correlations of EEG data. The input $x_{in} \in \mathbb{R}^{C \times T}$ is linearly transformed into equal-sized with weight metrics $W_q, W_k, W_v$:

$$x'_{in} = \text{Softmax}\left(\frac{W_q x_{in} \cdot (W_k x_{in})^T}{\sqrt{d}}\right) \cdot W_v x_{in} \tag{1}$$

where $x'_{in} \in \mathbb{R}^{C \times T}$ denotes the output after the SA module, $d$, the time length of EEG data, is a scaled factor to accommodate the Softmax function.

#### 3.3.2 GRAPH ATTENTION

We employ the GA module to update each electrode with all the others, using the implementation from Brody et al. (2022). We treat each electrode as a node $n_i \in \mathbb{R}^{1 \times T}$, $i = 1, ..., ch$, which has edges to all the other electrodes $\mathcal{N}_i$. The process to update one electrode is as follows:

$$n'_i = \alpha_{i,i} W n_i + \sum_{j \in \mathcal{N}_i} \alpha_{i,j} W n_j \tag{2}$$

where $n'_i$ denotes each node after processing, $\alpha_{i,j}$ is attention coefficients indicating importance of node $j$ features to node $i$, and $W$ is the weight of linear transformation. Calculate $\alpha_{i,j}$ as:

$$\alpha_{i,j} = \frac{\exp(a^T \text{LeakyReLU}(W[n_i \parallel n_j]))}{\sum_{k \in \mathcal{N}_i \cup \{i\}} \exp(a^T \text{LeakyReLU}(W[n_i \parallel n_k]))} \tag{3}$$

where $a \in \mathbb{R}^{2T}$ denotes the weight of a feedforward layer for attention calculation, $()^T$ is the transposition operator, and $\parallel$ is the concatenation operator. LeakyReLU with a slope of 0.2 is used for nonlinearity in calculation. Residual connection is employed by integrating the input and output of the SA and GA modules separately, contributing to stable training (He et al., 2016).

## 4 EXPERIMENTS AND RESULTS

### 4.1 DATASETS AND PREPROCESSING

The dataset (Gifford et al., 2022) contains EEG data from ten participants with a time-efficient rapid serial visual presentation (RSVP) paradigm. The training set includes 1654 concepts×10 images×4 repetitions. The test set includes 200 concepts×1 image×80 repetitions. Images for training and testing appear in a pseudo-randomized order, and a target image is used to reduce eye blinks and other artifacts. Each image displays 100 ms, followed by a 100 ms blank screen. Raw EEG data filtered to [0.1, 100] Hz has 63 channels and a sample rate of 1000 Hz.

For preprocessing, we epoched EEG data into trials ranging from 0 to 1000 ms after stimuli onset. Baseline correction was performed with the mean of 200 ms pre-stimulus data. All electrodes were preserved for analysis with down-sampling to 250 Hz, and multivariate noise normalization was performed with training data (Guggenmos et al., 2018). We averaged all EEG repetitions of one image to ensure the signal-to-noise ratio and compared the impact of repetitions. Images were resized to 224×224 and normalized before being processed by the image encoder.

### 4.2 EXPERIMENT DETAILS

Our method is implemented with PyTorch on a Geforce 4090 GPU. We randomly select 740 trials from training data as the validation set in each run of the code. Best models are saved when the validation loss reaches a minimum of 200 epochs in the training process. We compute the results of the test set once after training. It takes about 5 minutes per subject to train with a batch size of 1000, since we get the image features in advance. The $k$ in TSConv is set to 40, $m_1$ to 25, $m_2$ to 51, and $s$ to 5 by pre-experiments. Adam optimizer is used with the learning rate, $\beta_1$ and $\beta_2$ of 0.0002, 0.5, and 0.999, respectively. Wilcoxon Signed-Rank Test is employed to analyze the statistical significance.

Table 2: Overall accuracy (%) of 200-way zero-shot classification: top-1 and top-5

| Method | Subject 1 | | Subject 2 | | Subject 3 | | Subject 4 | | Subject 5 | | Subject 6 | | Subject 7 | | Subject 8 | | Subject 9 | | Subject 10 | | Ave | |
|---|---|---|---|---|---|---|---|---|---|---|---|---|---|---|---|---|---|---|---|---|---|---|
| | top-1 | top-5 | top-1 | top-5 | top-1 | top-5 | top-1 | top-5 | top-1 | top-5 | top-1 | top-5 | top-1 | top-5 | top-1 | top-5 | top-1 | top-5 | top-1 | top-5 | top-1 | top-5 |
| Subject dependent - train and test on one subject | | | | | | | | | | | | | | | | | | | | | | |
| BraVL | 6.1 | 17.9 | 4.9 | 14.9 | 5.6 | 17.4 | 5.0 | 15.1 | 4.0 | 13.4 | 6.0 | 18.2 | 6.5 | 20.4 | 8.8 | 23.7 | 4.3 | 14.0 | 7.0 | 19.7 | 5.8 | 17.5 |
| NICE | 12.3 | 36.6 | 10.4 | 33.9 | 13.1 | 39.0 | 16.4 | 47.0 | 8.0 | 26.9 | 14.1 | 40.6 | 15.2 | 42.1 | 20.0 | 49.9 | 13.3 | 37.1 | 14.9 | 41.9 | 13.8 | 39.5 |
| NICE - SA | 13.3 | 40.2 | 12.1 | 36.1 | 15.3 | 39.6 | 15.9 | 49.0 | 9.8 | 34.4 | 14.2 | 42.4 | 17.9 | 43.6 | 18.2 | 50.2 | 14.4 | 38.7 | 16.0 | 42.8 | 14.7 | 41.7 |
| NICE - GA | 15.2 | 40.1 | 13.9 | 40.1 | 14.7 | 42.7 | 17.6 | 48.9 | 9.0 | 29.7 | 16.4 | 44.4 | 14.9 | 43.1 | 20.3 | 52.1 | 14.1 | 39.7 | 19.6 | 46.7 | 15.6 | 42.8 |
| Subject independent - leave one subject out for test | | | | | | | | | | | | | | | | | | | | | | |
| BraVL | 2.3 | 8.0 | 1.5 | 6.3 | 1.4 | 5.9 | 1.7 | 6.7 | 1.5 | 5.6 | 1.8 | 7.2 | 2.1 | 8.1 | 2.2 | 7.6 | 1.6 | 6.4 | 2.3 | 8.5 | 1.8 | 7.0 |
| NICE | 7.6 | 22.8 | 5.9 | 20.5 | 6.0 | 22.3 | 6.3 | 20.7 | 4.4 | 18.3 | 5.6 | 22.2 | 5.6 | 19.7 | 6.3 | 22.0 | 5.7 | 17.6 | 8.4 | 28.3 | 6.2 | 21.4 |
| NICE - SA | 7.0 | 22.6 | 6.6 | 23.2 | 7.5 | 23.7 | 5.4 | 21.4 | 6.4 | 22.2 | 7.5 | 22.5 | 3.8 | 19.1 | 8.5 | 24.4 | 7.4 | 22.3 | 9.8 | 29.6 | 7.0 | 23.1 |
| NICE - GA | 5.9 | 21.4 | 6.4 | 22.7 | 5.5 | 20.1 | 6.1 | 21.0 | 4.7 | 19.5 | 6.2 | 22.5 | 5.9 | 19.1 | 7.3 | 25.3 | 4.8 | 18.3 | 6.2 | 26.3 | 5.9 | 21.6 |

Table 3: Classification accuracy (%) with different EEG encoder and Image encoder: top-1 (top-5)

| Method | Subject 1 | Subject 2 | Subject 3 | Subject 4 | Subject 5 | Subject 6 | Subject 7 | Subject 8 | Subject 9 | Subject 10 | Ave | std |
|---|---|---|---|---|---|---|---|---|---|---|---|---|
| EEG encoder | | | | | | | | | | | | |
| ShallowNet | 6.0 (16.0) | 3.5 (21.5) | 8.5 (28.0) | 16.0 (41.0) | 4.0 (16.5) | 9.5 (32.0) | 13.0 (31.0) | 10.0 (24.5) | 2.5 (9.0) | 6.0 (22.5) | 7.9 (24.2) | 4.3 (9.3) |
| DeepNet | 12.0 (31.0) | 7.5 (27.5) | 10.5 (33.0) | 14.0 (36.5) | 4.5 (20.0) | 10.0 (33.0) | 9.5 (30.0) | 12.0 (41.0) | 9.0 (31.5) | 10.0 (33.5) | 9.9 (31.7) | 2.6 (5.5) |
| Conformer | 11.0 (39.0) | 8.5 (28.0) | 11.0 (33.5) | 14.5 (38.0) | 6.0 (25.5) | 10.0 (30.0) | 13.0 (40.0) | 13.0 (35.5) | 10.5 (31.0) | 13.5 (37.5) | 11.1 (33.8) | 2.6 (5.0) |
| EEGNet | 11.5 (38.0) | 11.0 (35.5) | 14.5 (37.5) | 15.0 (43.5) | 11.0 (30.0) | 13.0 (42.5) | 14.0 (35.5) | 15.0 (42.5) | 11.0 (37.0) | 14.5 (41.5) | 13.1 (38.4) | 1.8 (4.2) |
| TSConv | 12.3 (36.6) | 10.4 (33.9) | 13.1 (39.0) | 16.4 (47.0) | 8.0 (26.9) | 14.1 (40.6) | 15.2 (42.1) | 20.0 (49.9) | 13.3 (37.1) | 14.9 (41.9) | 13.8 (39.5) | 3.3 (6.5) |
| Image encoder | | | | | | | | | | | | |
| ResNet | 9.5 (20.0) | 7.0 (11.5) | 7.5 (16.0) | 10.0 (18.0) | 6.5 (15.0) | 8.0 (14.0) | 10.0 (20.0) | 14.0 (25.0) | 6.0 (15.0) | 9.0 (18.5) | 8.8 (17.3) | 2.2 (3.6) |
| ResNet* | 8.0 (13.0) | 6.0 (11.0) | 6.0 (11.5) | 5.0 (9.5) | 4.5 (10.0) | 8.5 (12.0) | 6.0 (10.5) | 10.0 (14.5) | 6.0 (12.0) | 9.0 (13.5) | 6.9 (11.8) | 1.8 (1.6) |
| ViT | 8.5 (18.0) | 5.5 (15.0) | 10.5 (20.5) | 7.0 (20.5) | 6.0 (15.0) | 8.0 (20.5) | 7.5 (19.5) | 11.5 (21.5) | 7.0 (17.0) | 7.5 (17.5) | 7.9 (18.5) | 1.8 (2.2) |
| ViT* | 13.5 (27.0) | 13.0 (28.5) | 13.0 (29.0) | 13,5 (32.5) | 9.5 (21.5) | 12.5 (29.0) | 10.0 (34.0) | 18.0 (38.5) | 6.5 (24.5) | 11.0 (29.5) | 12.1 (29.4) | 3.1 (4.8) |
| CLIP | 12.3 (36.6) | 10.4 (33.9) | 13.1 (39.0) | 16.4 (47.0) | 8.0 (26.9) | 14.1 (40.6) | 15.2 (42.1) | 20.0 (49.9) | 13.3 (37.1) | 14.9 (41.9) | 13.8 (39.5) | 3.3 (6.5) |

* pre-trained ResNet-50 and ViT-B/16 models.

## 4.3 OVERALL PERFORMANCE

The decoding results are outlined in Table 2, with a sanity check in Appendix A.1. We evaluated the base framework (NICE) alongside its variants enhanced with self-attention (NICE-SA) and graph attention (NICE-GA). The evaluation was on the latest and most extensive image-EEG dataset, with only one state-of-the-art BraVL (Du et al., 2023) for comparison. This dataset posed a 200-way zero-shot task, comprising 200 untrained image concepts, with a chance level of 0.5%. In subject-dependent experiments, NICE achieved a top-1 accuracy of 13.8% and a top-5 accuracy of 39.5%, surpassing BraVL by 8.0% and 22.0%, respectively. The introduction of SA and GA improved the top-1 by 0.9% ($p < 0.05$) and 1.8% ($p < 0.01$), separately. Besides, in subject-independent experiments, our method achieved top-1 of 6.2% and top-5 of 21.4%. NICE-SA improved the top-1 by 0.8% ($p > 0.05$), while NICE-GA only improved for several subjects, with the average top-1 decreasing by 0.3%. We employed base NICE for fair comparison in the following experiments.

## 4.4 ENCODER COMPARISON

The comparison of EEG encoders and image encoders is present in Table 3. We carefully selected several representative methods, including ShallowNet, DeepNet (Schirrmeister et al., 2017), Conformer (Song et al., 2023), and EEGNet (Lawhern et al., 2018). Our custom-designed TSConv, which utilizes temporal and spatial convolution, outperformed these methods. The average top-1 accuracy of TSConv is 5.9% higher than ShallowNet ($p < 0.001$), 3.9% higher than DeepNet ($p < 0.001$), 2.7% higher than Conformer ($p < 0.001$), and 0.7% higher than EEGNet ($p > 0.05$). The robustness of TSConv reflected by standard deviation was not superior.

We utilized image encoders, including ResNet-50 pre-trained on ImageNet-1k, ViT-B/16 pre-trained on ImageNet-21k and fine-tuned on ImageNet-1k, and CLIP-ViT-L/14 pre-trained on 400 million image-text pairs. CLIP pre-trained on huge amounts of data helped us achieve the highest results. Interestingly, ViT also worked well with an average top-1 accuracy 1.7% lower than CLIP ($p > 0.05$). The results of ResNet were 6.9% lower ($p < 0.001$). See Appendix A.2 for the latest update.

We also showed the impact of pre-training compared with non-pre-trained ResNet-50 and ViT-B/16. Pre-trained ViT outperformed the non-pre-trained by 4.2% ($p < 0.01$). Surprisingly, ResNet pre-trained with smaller ImageNet-1k decreased by 1.9% ($p < 0.05$). Note that non-pre-trained models performed well but incurred substantially higher computational costs, with details in Appendix A.3.

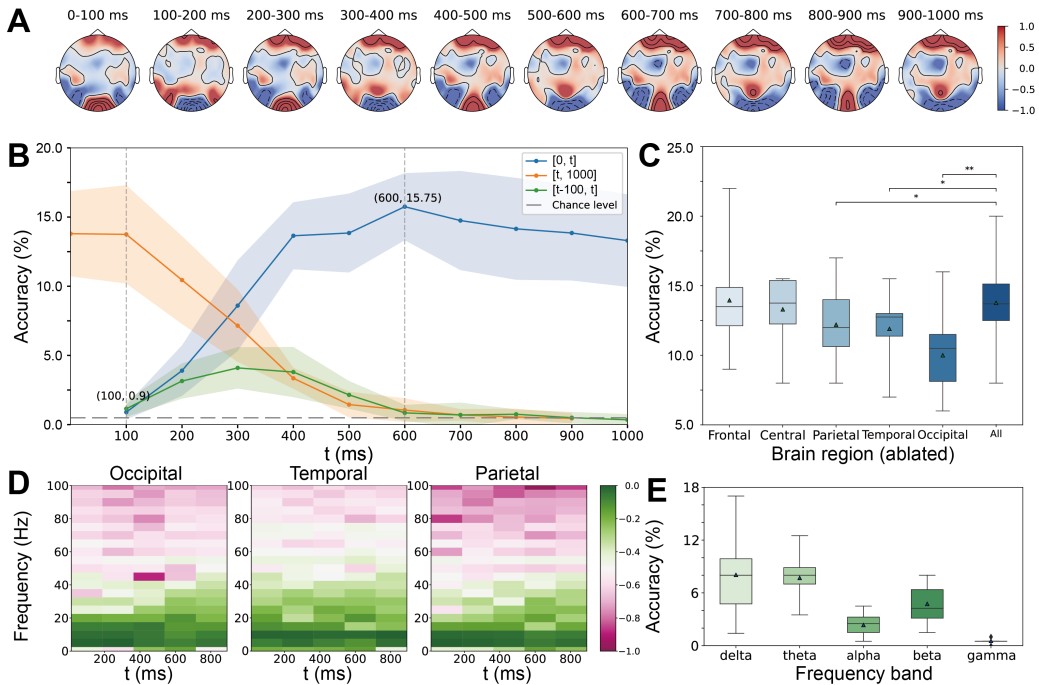

Figure 2: Temporal, spatial, spectral analysis. (A) Topographies of each 100 ms by averaging all training trials. The temporal lobe has a clear response between 100-600 ms. (B) Averaged accuracy of all subjects with different time lengths. The region of interest is 100-600 ms, after hysteresis in visual systems. (C) Ablate electrodes of different brain regions. The occipital, temporal, and parietal lobes contribute significantly to image decoding. (D) Time-frequency maps of the occipital, temporal, and parietal lobes data from one subject. The main components are below 30 Hz, and high-frequency components can be observed on the temporal lobe. (E) Averaged accuracy of different rhythms. Theta (∼4 Hz) and beta (∼14-18 Hz) bands show effective performance.

## 4.5 TEMPORAL, SPATIAL, AND SPECTRAL DYNAMICS

Beyond the self-supervised framework, we try to demonstrate the biological plausibility by resolving the visual processing of EEG signals. We conducted a detailed analysis from the temporal, spatial, and spectral dynamics perspective in Fig. 2. The topographies were first plotted with each 100 ms in Fig. 2(A) by averaging all training trials from the first subject. Visual masking (Keysers & Perrett, 2002) would be alleviated because the data used were collected in many rapid series sequences. A clear response could be observed on the temporal cortex 100-600 ms after the onset, although the 200 ms stimulus onset asynchrony (SOA) still caused periodic responses on the occipital cortex. Besides, the parietal cortex also had a response after 100 ms. The phenomenon is consistent with the bottom-up hierarchy of visual system (DiCarlo & Cox, 2007), that the visual stimulus is processed sequentially by the V1, V2, V4 on the occipital cortex, and inferotemporal (IT) on the temporal cortex along the ventral stream for object recognition (Bao et al., 2020).

We explored the active time range in three ways: incrementing forward, decrementing backward, and segmentation, as in Fig. 2(B). It could be seen that the region of interest was in 100 to 600 ms. The average top-1 accuracy of [0, 1000] ms was 0.1% lower than that of [100, 1000] ms without significance ($p > 0.05$). These findings suggest that the initial 100 ms following stimulus onset contained limited information, possibly due to the hysteresis effect in the visual pathway (Sayal et al., 2020). Besides, the signal after 600 ms brought a negative effect, probably due to the noise from other stimuli and cognitive processing, emerging a response increasing on the frontal lobe as in Fig. 2(A).

We conducted an ablation study involving electrodes from different cortices as in Fig. 2(C). The average accuracy sharply declined 3.8% ($p < 0.01$) without occipital electrodes. Ablation of temporal,

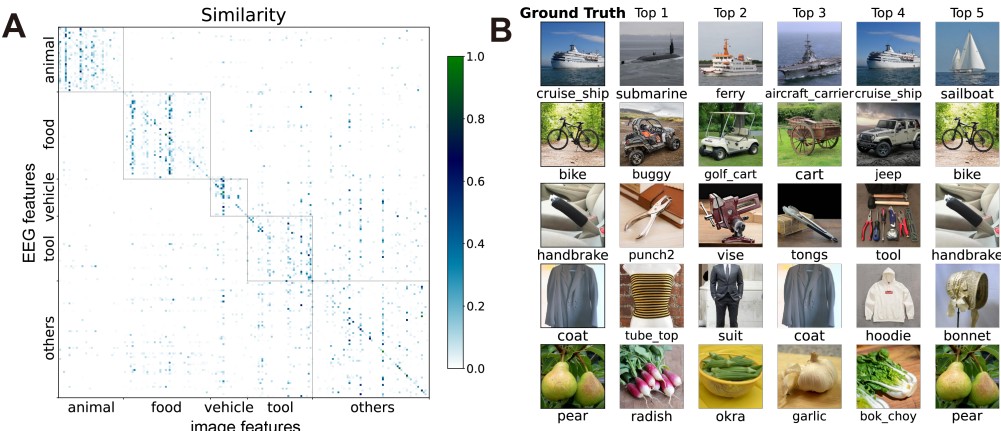

Figure 3: Semantic similarity analysis and visualization. (A) Cosine similarity of feature pairs of 200 concepts in the test set. The results calculated by the trained models of 10 subjects were averaged, and all the concepts were rearranged into five categories: animal, food, vehicle, tool, and others. (B) Classification results visualized with ground truth (first column) and the top-5 predicted.

parietal electrodes and the central area near the motor cortex decreased the results by 1.9% ($p < 0.05$), 1.6% ($p < 0.05$), and 0.5% ($p > 0.05$) separately. The accuracy improved by 0.2% ($p > 0.05$) to discard frontal electrodes. The results are still reasonable with the low spatial resolution of EEG.

We plotted time-frequency maps in Fig. 2(D) by averaging all training trials from the first subject. The main components were below 30 Hz from the occipital, temporal, and parietal region data. High-frequency responses could be observed from electrodes on the temporal cortex. Interestingly, the frequency responses had an upward trend along with visual processing.

We further conducted classification tests with different frequency bands in Fig. 2(E). Theta and delta bands near 4 Hz performed well, where theta was more stable for different subjects. The beta and alpha bands also demonstrated performance above the chance level, coarsely aligning with previous findings (Bastos et al., 2015; Michalareas et al., 2016). However, the gamma band could hardly help us in decoding, for which we have two speculations. Capturing pure gamma oscillations from EEG can be challenging due to the susceptibility of high-frequency artifacts (Fries et al., 2008; Yuval-Greenberg et al., 2008). Besides, the amplitude of the gamma band is quite low and easily modulated by other cognitive processing, such as attention and memory (Herrmann & Demiralp, 2005). This may explain why some studies with MEG have also disregarded the analysis of the gamma band (Hebart et al., 2023). Inspired by the latest work Benchetrit et al. (2023), we provided preliminary analysis on MEG data for reference in Appendix A.6.

## 4.6 SEMANTIC SIMILARITY

We show the semantic similarity and classification visualization in Fig. 3. One concern is that the discrimination we obtain from EEG is not due to semantic information but only the basic visual components, such as color, brightness, contrast, etc. We performed representational similarity analysis (RSA) (Cichy & Oliva, 2020), and got the similarity matrices by averaging all subjects, and categorizing the 200 concepts in the test set into animal, food, vehicle, tool, and others. As shown in Fig. 3(A), we could observe distinct intra-category aggregation. This meant the obtained EEG representations were closer to the image representations within the corresponding semantic category.

The images in the test set were used for visualization in Fig. 3(B). We randomly chose several top-5 decoding results from the first subject. It could be noticed that the predicted results were semantically similar to the ground truth, such as the bike was predicted into several vehicles with wheels.

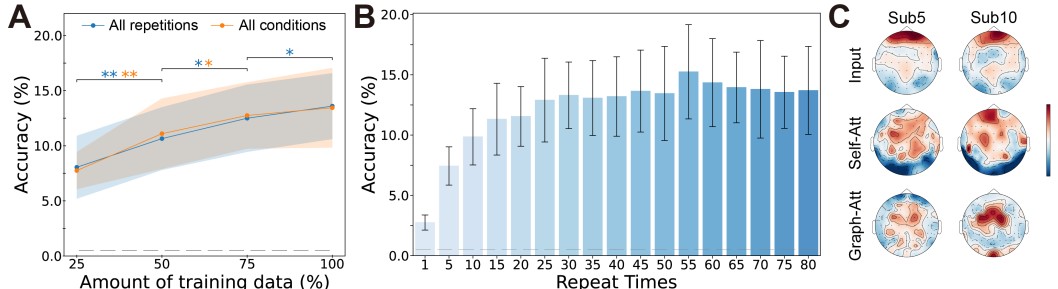

Figure 4: Effect of data size and repetition in training and test set, and visualization of SA and GA. (A) Accuracy with quarters of conditions and all repetitions, with quarters of repetitions and all conditions of training images. Adding more conditions can potentially further improve the performance. (B) Accuracy with different repetitions of the test images. Average ten times to achieve an accuracy of 9.9 (30.1)%. (C) Grad-CAMs of SA and GA show activation on temporal and occipital regions.

### 4.7 DATA SIZE AND REPETITION

We compared the impact of the data size and repetition of the training and test sets, which may be crucial for the performance, in Fig. 4. There were 1654 concepts×10 image conditions repeated four times in the training set. We averaged the four repetitions of each condition for training. Two cases were compared in Fig. 4(A), adding conditions by 25% intervals with all repetitions, and adding repetitions by 25% with all conditions. Increasing conditions, i.e., data size, contributed significantly to decoding accuracy from 25% to 50% ($p < 0.01$), from 50% to 75% ($p < 0.05$), and from 75% to 100% ($p < 0.05$). There were also significant contributions by increasing repetitions from 25% to 50% ($p < 0.01$), from 50% to 75% ($p < 0.05$), except from 75% to 100% ($p > 0.05$). The results suggest that increasing repetitions is of limited help, while more data promises further improvements.

From another view, repetitions of test data determine the practice efficiency. We compared the number of repetitions for averaging, from 5 to 80, with an interval of 5 in Fig. 4(B). Ten repetitions achieved a 9.9% (30.1%) accuracy, which tended to stabilize above 13.0% (37.0%) after twenty-five repetitions.

### 4.8 EFFECT OF SPATIAL MODULES

We tried to provide additional evidence of brain responses facilitated by spatial modules. Gradient-weighted class activation mapping (Grad-CAM) (Selvaraju et al., 2017) was used to visualize the interest region of SA and GA in Fig. 4(C) and Appendix A.4. The original data was largely affected by the response on the frontal lobe. Surprisingly, self and graph attention focuses on the temporal and occipital lobes, providing implicit evidence that object recognition from EEG is a feasible endeavor.

## 5 DISCUSSION AND CONCLUSION

There have been some studies achieving commendable results in image reconstruction with fMRI. We turn to more convenient and fast EEG signals and focus on the object recognition tasks, in which the semantic information is the significant gain by natural image decoding compared to visual decoding of contrast, color, etc. We have tried to demonstrate the feasibility and plausibility of EEG-based image decoding from three folds, zero-shot classification performance, detailed resolving of the brain activity, and model interpreting. There are also some limitations of this paper. Firstly, we observed that a stable response required multiple repetitions. This could be attributed to the brief 100 ms stimulus duration, which may make it susceptible to being missed or disrupted by stimuli before and after. We will aim to identify a more optimal window length for stimulus presentation. Secondly, we have yet to capture useful information from the gamma band, which deserves further investigation.

In conclusion, we propose a self-supervised framework to decode natural images from EEG for object recognition. Our framework has achieved remarkable results in zero-shot tasks with rich biological evidence. The results provide new inspiration for practical brain-computer interfaces.

ACKNOWLEDGMENTS

This work was supported in part by the National Natural Science Foundation of China under Grant U2241208, and the Key Research and Development Program of Ningxia under Grant 2023BEG02063.

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

# A APPENDIX

## A.1 SANITY CHECK

In this work, we perform 200-way zero-shot tasks. Here, we provide an experimental chance by shuffling pairs to compare with the ideal chance of 0.5% in Table 4. The experimental chance is close to the ideal chance, and is significantly lower than the performance of the NICE model ($p < 0.001$).

Table 4: Chance level of the experiments

| Condition | Top-1 (%) | Top-5 (%) |
|---|---|---|
| ideal chance | 0.5 | 2.5 |
| experimental chance | 0.3 | 2.3 |
| NICE | 13.8 | 39.5 |

## A.2 BETTER ENCODERS

The adaptability of the NICE framework extends to any other well-designed EEG or image encoders, as shown in Table 3. We further present results utilizing EVA-CLIP Sun et al. (2023) pre-trained with LAION-2B as the image encoder in Table 5. The enhancements in both overall top-1 ($p < 0.001$) and top-5 ($p < 0.01$) accuracy are surprising. The results further show the potential of this task and hint at the consistency of large pre-trained models and brain responses. Beyond that, we recommend not solely focusing on overall performance, but the specific insight gained.

Table 5: Overall accuracy (%) with EVA-CLIP image encoder: top-1 and top-5

| | Subject 1 | | Subject 2 | | Subject 3 | | Subject 4 | | Subject 5 | | Subject 6 | | Subject 7 | | Subject 8 | | Subject 9 | | Subject 10 | | Ave | |
|---|---|---|---|---|---|---|---|---|---|---|---|---|---|---|---|---|---|---|---|---|---|---|
| Method | top-1 | top-5 | top-1 | top-5 | top-1 | top-5 | top-1 | top-5 | top-1 | top-5 | top-1 | top-5 | top-1 | top-5 | top-1 | top-5 | top-1 | top-5 | top-1 | top-5 | top-1 | top-5 |
| **NICE** | 17.8 | 44.2 | 14.6 | 39.4 | 19.1 | 48.9 | 20.8 | 52.2 | 11.3 | 36.9 | 19.1 | 48.7 | 21.0 | 49.5 | 26.9 | 59.5 | 16.0 | 45.2 | 20.3 | 51.7 | 18.7 | 47.6 |
| **NICE - SA** | 17.1 | 45.4 | 15.3 | 44.2 | 17.4 | 49.0 | 22.1 | 52.0 | 15.6 | 42.4 | 21.1 | 50.5 | 23.4 | 52.5 | 26.3 | 58.8 | 18.3 | 49.1 | 22.1 | 53.5 | 19.9 | 49.7 |
| **NICE - GA** | 19.8 | 46.7 | 19.6 | 44.7 | 19.8 | 51.5 | 25.7 | 54.5 | 13.2 | 39.2 | 23.4 | 54.1 | 22.7 | 52.9 | 26.8 | 58.9 | 17.8 | 47.1 | 23.6 | 53.9 | 21.2 | 50.4 |

## A.3 COMPARISON OF PRE-TRAINING

We employed pre-trained models as the image encoder for two reasons: 1) Pre-trained models like CLIP and ViT possess extensive feature extraction capabilities gained from large-scale training. We sought to leverage these models to enhance the EEG model, trained on a smaller dataset. 2) Training the image encoder demands substantial computational resources. In our current pre-trained setup, one GPU with 8GB memory yields a well-trained EEG encoder for one subject in 5 minutes.

We conducted additional experiments without pre-trained image encoders in Table 6, training the randomly initialed image encoder and the EEG encoder simultaneously. ViT-B/16 and ResNet-50 were used as examples. CLIP also relies on ViT-L, but with more layers than ViT-B (24 vs. 12), impeding completion. In the scenario with non-pre-trained, we observed that although training loss reduced rapidly, validation loss demonstrated a weaker decline, which is a sign of limited generalization. During testing, pre-trained ViT outperformed ($p < 0.01$) non-pre-trained models. Pre-trained ResNet decreased by 1.9% ($p < 0.05$) instead, hinting at avenues of refining the image encoder to enhance overall performance. But still, training the image encoder imposes a significant computational load. We should mention that the decrease in computational cost actually comes from the freezing of the pre-trained image encoder and getting image features beforehand.

## A.4 SPATIAL MODULE INTERPRETATION

We provide Grad-CAMs and attention weights of all subjects in Fig. 5. These visualizations show the alleviation of prefrontal region influence—often tied to cognitive behavior like workload and memory—by both SA and GA. Intriguingly, GA focuses on the occipital region, integral to visual processing, across multiple subjects. SA appears adept at capturing temporal region activities

Table 6: Comparison of pre-trained and non-pre-trained image encoders

| Image Enc | Top-1 (%) | Top-5 (%) | Params. | Memory | Time per subject |
|---|---|---|---|---|---|
| ViT | 7.9 | 18.5 | 88.0 M | ∼154 G | ∼ 130 min |
| ViT* (frozen) | 12.1 | 29.4 | 1.8 M | ∼8 G | ∼5 min |
| ResNet | 8.8 | 17.3 | 25.3 M | ∼112 G | ∼60 min |
| ResNet* (frozen) | 6.9 | 11.8 | 1.8 M | ∼8 G | ∼5 min |

[*] pre-trained ResNet-50 and ViT-B/16. The image features were obtained before training.

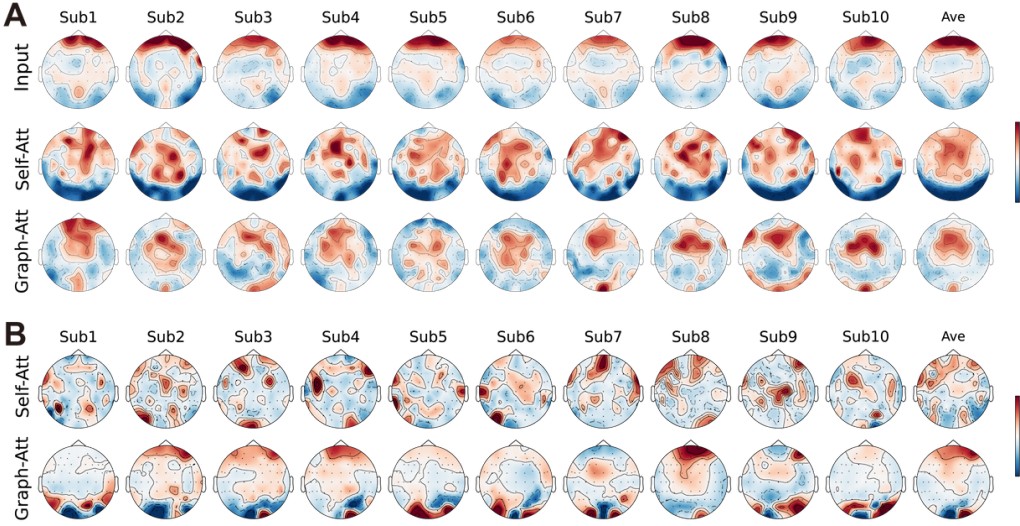

Figure 5: (A) Grad-CAM for self-attention (SA) and graph attention (GA) modules of individual subjects and the average across subjects. (B) Attention weights of SA and GA with individual subjects and the average across subjects. The visualizations highlight the regions of interest, focusing on the temporal and occipital brain areas, which are known to be associated with visual processing.

associated with high-level visual processing. Our model interpretation demonstrates that EEG responses can capture and reveal brain activity in the occipital and temporal areas, which are strongly associated with object recognition processes. This provides implicit evidence for the effectiveness of EEG-based object recognition, and further validates the plausibility of our approach.

## A.5 PARAMETER SENSITIVITY

The temperature parameter ($\tau$) in contrastive learning plays a crucial role in shaping the similarity distribution. In our implementation, we optimized this temperature directly as a log-parameterized multiplicative scalar during training, similar to the CLIP model. To explore the impact of temperature settings, we conducted a series of comparisons, and the results are presented in Table 7. These comparisons demonstrate the significant influence of temperature adjustments on the results. Besides, our pre-experiments help determine some conventional parameters such as batch size, decay, etc.

Table 7: Comparison of the temperature parameter in contrastive learning

| Temperature | learnable | 0.001 | 0.005 | 0.01 | 0.03 | 0.05 | 0.1 | 0.5 |
|---|---|---|---|---|---|---|---|---|
| Top-1 (%) | 13.8 | 12.4 | 13.45 | 15.8 | 15.2 | 14.7 | 12.1 | 10.5 |
| signif. (p) | - | >.05 | >.05 | <.01 | <.01 | <.05 | <.01 | <.001 |

## A.6 MEG EXPERIMENTS

We provided analysis on an additional MEG dataset Hebart et al. (2023) to corroborate the results on EEG data. The MEG dataset of four participants has 271 channels and more stable responses with a long stimulus duration of 500 ms followed by a blank screen of 1000±200 ms. There are 1854 concepts×12 images×1 repetitions in the training stage and 200 concepts×1 image×12 repetitions in the test stage. We discarded 200 test concepts from the training set to construct the zero-shot task. The MEG data were epoched into trials from 0 to 1000 ms after the stimuli onset. We used a band-pass filter of [0.1, 100] Hz and baseline correction after down-sampling to 200 Hz for preprocessing. We averaged all MEG repetitions of one image to ensure the signal-to-noise ratio. Note that the statistical analysis was not performed on the MEG dataset due to the few participants.

Table 8: Overall accuracy (%) of 200-way zero-shot classification and retrieval on MEG data

| Task | Method | Sub 1 | | Sub 2 | | Sub 3 | | Sub 4 | | Ave | |
|---|---|---|---|---|---|---|---|---|---|---|---|
| | | Top-1 | Top-5 | Top-1 | Top-5 | Top-1 | Top-5 | Top-1 | Top-5 | Top-1 | Top-5 |
| classification | NICE | 6.9 | 20.5 | 15.3 | 37.1 | 12.3 | 35.0 | 5.8 | 21.1 | **10.1** | **28.4** |
| | w/ SA | 7.3 | 22.6 | 13.1 | 37.1 | 10.2 | 30.2 | 8.3 | 24.5 | 9.7 | 28.6 |
| | w/ GA | 6.4 | 23.4 | 16.2 | 40.8 | 14.0 | 38.9 | 6.4 | 25.2 | 10.8 | 32.1 |
| retrieval | NICE | 9.6 | 27.8 | 18.5 | 47.8 | 14.2 | 41.6 | 9.0 | 26.6 | **12.8** | **36.0** |
| | w/ SA | 9.8 | 28.1 | 18.6 | 46.4 | 10.5 | 38.4 | 11.7 | 27.2 | 12.7 | 35.0 |
| | w/ GA | 8.7 | 30.5 | 21.8 | 56.6 | 16.5 | 49.7 | 10.3 | 32.3 | 14.3 | 42.3 |

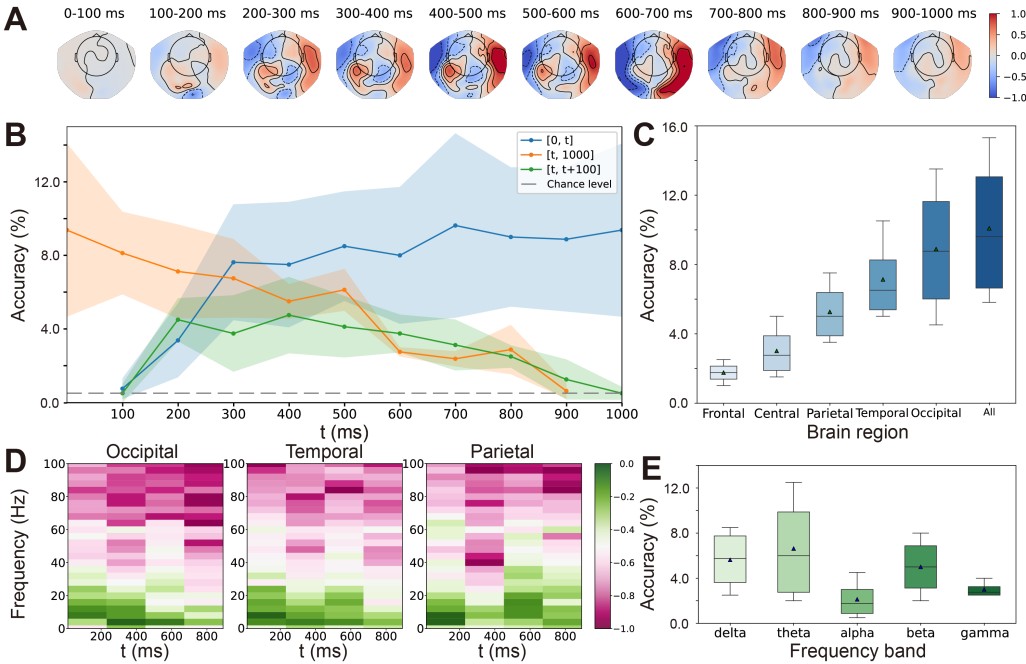

Figure 6: Temporal, spatial, spectral analysis of MEG data. (A) Topographies of each 100 ms by averaging training trials. (B) Averaged accuracy with different time lengths. (C) Averaged accuracy of different brain regions (not ablation here for clarity). (D) Time-frequency maps of the occipital, temporal, and parietal data from one subject. (E) Averaged accuracy of different rhythms.

Classification and retrieval tasks with the MEG dataset have been conducted with the NICE framework in Table 8. A similar 200-way zero-shot classification showed above-chance performance on MEG data with an average top-1 accuracy of 10.1% and top-5 accuracy of 28.4%. Besides, we provided the results on a retrieval task, which meant directly using the stimulus images for matching templates instead of other images belonging to the concepts. The spatial module GA could still help with the

results, but SA varied across individuals. It is worth noting that a more suitable encoder for extracting MEG features should help with the performance Benchetrit et al. (2023). Moreover, we roughly conducted the temporal, spatial, and spectral analysis of MEG data in Fig. 6. The results were similar to Fig. 2, while the temporal responses changed because of the longer stimuli duration.

## A.7 EEG Encoder Comparison

The residual connection He et al. (2016) was used with SA and GA modules to stabilize the training procedure. We compared the effect of the residual connection in Table 9. It can be seen that it significantly helped GA in top-1 accuracy ($p<0.01$) but no significance for SA ($p > 0.05$).

The EEG encoder TSConv in our framework has a concise architecture consisting of basic temporal and spatial convolutional layers. We compared the details in Table 10 because the design was similar to ShallowNet Schirrmeister et al. (2017). The comparison encompassed several issues, including the kernel size, pooling order, and activation function. All three cases caused a significant effect on the top-1 accuracy compared to TSConv ($p < 0.01$). Besides, we provided the number of parameters and the output dimension within different EEG encoders in Table 11. We believe that other well-designed EEG feature extractors can help us improve the performance in this framework.

Table 9: Residual connection comparison for spatial modules

| Module | Res | Top-1 (std) | Top-5 (std) |
|---|---|---|---|
| SA | w/ res | 14.7 (2.3) | 41.7 (4.6) |
| | w/o res | 13.8 (2.8) | 39.6 (5.7) |
| GA | w/ res | 15.6 (2.9) | 42.8 (5.5) |
| | w/o res | 10.1 (3.2) | 29.9 (5.3) |

Table 10: Comparison of the architecture of TSConv

| Condition | Top-1 (std) | Top-5 (std) |
|---|---|---|
| original ShallowNet | 7.9 (4.3) | 24.2 (9.3) |
| kernel size as ShallowNet | 9.9 (2.4) | 33.5 (5.4) |
| pooling order | 10.9 (3.0) | 34.3 (7.2) |
| ELU->ReLU | 11.0 (2.5) | 33.3 (6.6) |
| TSConv | 13.8 (3.3) | 39.5 (6.5) |

## A.8 Features Distribution

We plotted t-SNE van der Maaten & Hinton (2008) in Fig. 7 to show the distribution variance between training and test sets with visual and EEG features separately, to be a supplement of the RSA in Fig. 3.

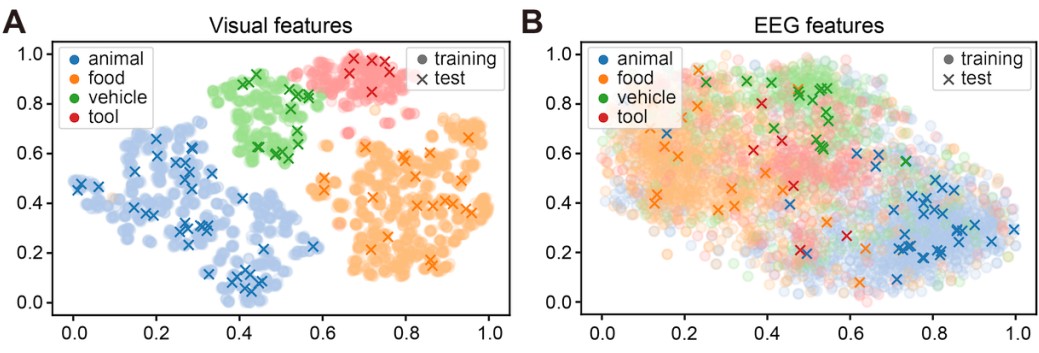

Figure 7: t-SNE visualization of four categories: animal, food, vehicle, and tool. (A) Visual feature distribution of the training and test sets. (B) EEG feature distribution of the training and test sets.

Table 11: Details of different EEG encoders

| EEG encoder | Params | Out dim |
|---|---|---|
| ShallowNet | 102.0 k | 440 |
| DeepNet | 303.3 k | 1400 |
| Conformer | 161.2 k | 1440 |
| EEGNet | 12.8 k | 1248 |
| TSConv | 102.0 k | 1440 |

Four categories, animal, food, vehicle, and tool, were selected for the visualization, which revealed why we could achieve zero-shot performance after training with rich data.

