# OpenReview forum: "Decoding Natural Images from EEG for Object Recognition"
_ICLR.cc/2024/Conference — ICLR 2024 poster_

### Official Review · Reviewer_pm39 · 2023-10-27

**Soundness:** 2 fair
**Presentation:** 3 good
**Contribution:** 2 fair
**Rating:** 3
**Confidence:** 4

**Summary:**

The authors propose a self-supervised framework: Natural Image Contrast EEG (NICE) to decode image from EEG

**Strengths:**

Clarity:
Describes the proposed NICE model with very clear logic and diagrams

**Weaknesses:**

Motivation:
The authors' main MOTIVATION consists in solving the following two problems (the problems mentioned in the second paragraph of the introduction are not sure whether they are motivations, so they are placed in the Question section): 1) Existing work has predominantly relied on supervised learning, often dealing with limited data from a few categories (introduction, first line of the third paragraph); 2) Limitations of the EEG feature extractor: convolution in the temporal and spatial dimensions respectively (the second sentence from the bottom of the third paragraph of introduction)
W1:Indeed most of the existing EEG tasks (emotion recognition, motor imagery) are supervised learning, but Decoding tasks (such as EEG2speech, EEG2Image) are different from traditional EEG tasks. The essence of the task is cross-modal conversion rather than Classification, self-supervised learning is a very common method in decoding tasks. Comparisons with supervised learning for traditional tasks are therefore unreasonable.
W2: Many existing EEG feature extraction methods focus on the spatial-temporal relationship of EEG, such as methods based on spatial-temporal graph, or spatial-temporal attention, or spatial-temporal convolution that is very similar to TSconv in NICE (Such as TSception). However, these very common methods are not mentioned in this paper, so this motivation is unreasonable.

Experiment：
W1: The main experiment only included one baseline, but there has been a lot of work on EEG vision decoding, such as Mind-Vis in CVPR. It is difficult to demonstrate the superiority of NICE by only comparing it with one baseline on a not commonly used dataset.
W2: There is no analysis of why NICE achieves better results than BraVL.
W3: In the Encoder comparison experiment, the baseline compared with TSConv did not include methods used for EEG feature extraction in the past five years.

**Questions:**

Q1: In the second paragraph of the Introduction, the author mentioned four problems with existing work. Are these problems the motivation of this paper? If so, why are some issues not mentioned or solved in the follow-up; if not, why are the issues that have not been addressed and paid attention to mentioned in the paper?
Q2: If so, for P3: “overlooking the inferior temporal cortex plays a necessary role in object recognition (fifth line)”, the inferior temporal cortex is located at the bottom of the frontal lobe, and EEG mainly records signals from the surface layer, so how does NICE pay attention to it from EEG and how the signals from the inferior temporal cortex can help NICE perform Vision Decoding?
Q3: For P4: "focuses on the peak of pairwise decoding about 110 ms after stimulus onset. (fourth line from the bottom)", the paper does not mention whether NICE's focus on the temporal dimension of the EEG signal is biologically interpretable, but P4 points out that the problem with the current model is that the focus of specific EEG time intervals is unreasonable

---

> ### Author Response · Authors · 2023-11-19
> **Response to Review pm39**
>
> Thanks for your comments to help us clarify the main contribution.
>
> Motivation
>
> Our core motivation is to prove the feasibility of EEG-based object recognition. On one hand, we proposed the self-supervised framework and achieved largely above-chance results on a rich dataset. On the other hand, we used the framework to resolve the temporal, spatial, and spectral dynamics of the brain activity to demonstrate the biological plausibility. The semantic representation and spatial module analysis gave further evidence. Thus, we think the main contribution of this work is not to design novel algorithms.
>
> W1. Is self-supervised learning new?
>
> A1. We know that self-supervised learning has been widely used in many fields. The limitations (few categories, supervised way) stated in the third paragraph of the Introduction were only with respect to the area of EEG-based image decoding. We want to use the most concise but effective way to solve the meaningful task.
>
> W2. Is EEG encoder new?
>
> The temporal-spatial convolution (TSConv) is a widely use way in EEG analysis, as mentioned in Sec. 3.1. One problem indeed is that most methods arrange EEG channels into one dimension, thus breaking the spatial characters. So here we introduced two modules, self-attention and channel attention (also well-known), to capture spatial features. Table 2 shows that the two tricks help the performance, therefore, we could use the grad-CAM and attention weights visualization in Fig. 4(C) and **Appendix A.3** Fig. 5 to reversely provide evidence of the brain responses we captured from EEG.
>
> Experiments
>
> W1. Why no more baseline was compared?
>
> A1. Most current image decoding works use fMRI, like mind-vis [R12]. However, fMRI and EEG are largely different in the acquisition and processing pipeline. fMRI relies on spatial resolution and commonly gets a 1-D vector of voxels after voxel selection or Principal Component Analysis. EEG relies on temporal resolution and gets time-series signals with multi-channels. Although there have been some methods using fMRI for this task, it’s hard to compare with EEG directly. Here, we would like to focus on EEG because it is low-cost, portable, and much faster than fMRI, which is more suitable for real-world applications. The dataset [R13] used in this work was published last December by a very reputable group in the field.
>
> W2. Why NICE is better than BraVL.
>
> A2. In our view, BraVL [R14] has good feature extraction but relatively basic Enc-Dec (MLP) and classifier (SVM). Besides, we think it’s a fMRI-oriented method due to more evaluation of two fMRI datasets. Thus, the performance on EEG is lower.
>
> W3. EEG encoder comparison.
>
> A3. We used several most recognized CNN-based EEG baseline methods for comparison [R15] [R16], though not the latest but still very effective. We also compared with the new method based on CNN+Transformer [R17].

---

> ### Author Response · Authors · 2023-11-19
> **Response to Review pm39**
>
> Q1. The issues in the second paragraph of the Introduction.
>
> A1. Thanks for the careful check. The second paragraph of the Introduction includes the main motivations, where there are several issues. (1) Previous works suffer from flawed experiment designs, single subject, and limited image categories. (2) The latest studies have not yet yielded sufficient multi-class decoding performance and plausibility, such as space and time analysis.
>
> Q2. If EEG can collect responses on the temporal cortex.
>
> A2. The inferior temporal (IT) cortex is reported to be related to human object recognition located below the middle of temporal areas. The standard 10-20 system of EEG electrode location includes temporal areas [R18]. We can capture electrical signals near IT, albeit attenuated by the skull.
> In Fig. 2(C), the overall performance significantly decreased (p < 0.05) when we ablating the temporal EEG channels. The results demonstrated that temporal information helped with the decoding task. We also observed response near the temporal region on topographies in Fig. 2(A). Besides, the grad-CAM and attention weight visualization in Fig. 4(C) and Appendix A.3 Fig. 5 show the tendency of the spatial modules to focus on the occipital and temporal regions.
>
> Q3. The plausibility of focusing on the temporal (time) dimension.
>
> A3. There are two folds for the interpretative evidence with the basis of some neuroscience research [R19] [R20]. (1) The beginning 100 ms after stimuli show non-significant results for the decoding task (p > 0.05) as in Fig. 2(B). We know this time latency mainly consists of the transmission of the visual pathway (retina, optic nerve, LGN) and a few processing of the primary visual cortex (V1) for low-level features. (2) The signal between 100-600 ms shows above-chance performance, especially 200-400 ms, aligning with the responses in Fig. 2(A). This latency consists of the responses along the ventral occipitotemporal (VOT) cortex, especially in the late stage of object recognition. This implies that we utilize not only low-level but also high-level information, which is the most important gain of using natural image stimuli instead of low-level stimuli such as contrast, color, etc.
>
> We believe the detailed resolving of temporal dynamics, as well as spatial and spectral aspects, will give reference to improve the performance and design of real-world applications.
>
> Thanks again! Look forward to any further feedback!
>
> [R12] Chen, Z., et al., Seeing beyond the brain: Conditional diffusion model with sparse masked modeling for vision decoding, CVPR, 2023. \
> [R13] Gifford, A. T., et al., A large and rich EEG dataset for modeling human visual object recognition, Neuroimage, 2022. \
> [R14] Du, C., et al., Decoding visual neural representations by multimodal learning of brain-visual-linguistic features. IEEE t-PAMI, 2023. \
> [R15] Schirrmeister, R. T., et al., Deep learning with convolutional neural networks for EEG decoding and visualization, HBM, 2017. \
> [R16] Lawhern, V. J., et al., EEGNet: a compact convolutional neural network for EEG-based brain–computer interfaces, JNE, 2018. \
> [R17] Song, Y., et al., Convolutional transformer for EEG decoding and visualization. IEEE t-NSRE, 2022. \
> [R18] https://en.wikipedia.org/wiki/10%E2%80%9320_system_(EEG) \
> [R19] Cichy, R. M., et al., Resolving human object recognition in space and time, Nature Neuroscience, 2014. \
> [R20] Xu, R., et al., A temporal hierarchy of object processing in human visual cortex, bioRxiv, 2023.

---

> ### Author Response · Authors · 2023-11-23
> **Response to Review pm39**
>
> Sorry to bother you. We are about to run out of time to respond.
>
> We have made complements and explanations for this work with the help of all the reviews, as summarized in the above 'General Response.' We would be grateful if you could confirm whether the rebuttal meets your expectations and if there is any other suggestion.
>
> Thanks anyway for your time and helpful comments!

---

### Official Review · Reviewer_9bdz · 2023-10-29

**Soundness:** 3 good
**Presentation:** 3 good
**Contribution:** 3 good
**Rating:** 8
**Confidence:** 5

**Summary:**

The work presents a method for the task of EEG-based object recognition, using a recent dataset collected with the Rapid Serial Visual Presentation (RSVP) paradigm. The authors propose a self-supervised framework where an image and an EEG encoder network are used to extract visual and EEG features. During training, the method maximizes the similarity between positive pairs of samples (i.e. when the signals of the EEG sample of a pair are obtained from the image sample of the pair), and minimizes the similarity for negative pairs. After training, the authors perform zero-shot EEG decoding, by matching EEG features to visual features from image categories that were not present during training. Experiments are performed in one dataset, using various network architectures for EEG and image feature extraction, presenting results that are superior to the previous state-of-the-art and demonstrating the importance of EEG electrode areas, EEG frequency bands, and the temporal window timings of the input EEG signals.

The contributions are:
1) an architecture for EEG feature extraction, named "TSConv", along with two plug-and-play attention modules that can be incorporated in the architecture
2) the training methodology of the proposed method, called "NICE" (Natural Image Contrast EEG)
3) the investigation of natural image information from EEG signals

**Strengths:**

There is some originality behind the work presented in the manuscript. The authors work on a recent RSVP dataset of EEG & visual data (Gifford et al., 2022), being among the first works that conduct experiments on the task of EEG-based object recognition using this dataset and presenting superior results compared to the previous state-of-the-art (Du et al., 2023). The contrastive loss employed during training, is different from contrastive losses that have been used in other works for EEG-based object recognition (e.g. in (Palazzo et al., 2021)).

Gifford et al., "A large and rich EEG dataset for modeling human visual object recognition", NeuroImage 2022
Du et al., "Decoding Visual Neural Representations by Multimodal Learning of Brain-Visual-Linguistic Features", TPAMI 2023
Palazzo et al., "Decoding Brain Representations by Multimodal Learning of Neural Activity and Visual Features", TPAMI 2021

The authors are aware of the challenges on the studied task, presenting a clear summary of previous works relevant to EEG-based object recognition. The presentation of the results provides clear insights for several factors that are involved (e.g. intra-subject versus cross-subject testing in Table 2, network architectures in Table 3, ablation studies on the attention modules in Table 2, electrode areas, EEG timings and EEG frequency bands in Figure 2).

**Weaknesses:**

The proposed architecture for EEG feature extraction, TSConv (described in Table 1 of the manuscript), is highly similar to that of the ShallowNet (as described in (Schirrmeister et al., 2017)) architecture, which the authors include as a baseline method. Specifically, the ShallowNet architecture originally consists of a temporal and a spatial convolutional layer, followed by average pooling and a fully-connected classification layer. Leaving aside the differences on the employed activation functions between TSConv and ShallowNet, one could say that the difference lies only on the order of placement for the average pooling layer. Given that the results reported in Table 3 are quite different for these two architectures (i.e. ShallowNet is the worst performing architecture and TSConv is the best performing architecture), what could be the interpretation?
Schirrmeister et al., "Deep learning with convolutional neural networks for EEG decoding and visualization", Human Brain Mapping 2017

Regarding the two plug-and-play spatial modules, the authors state that they are "instrumental in preserving the spatial characteristics of EEG channels" (page 3, Section 3.1). Considering the self-attention (SA) module and looking at Equation (1), one can notice the spatial filtering nature of the proposed self-attention. However the authors do not provide any insight on the spatial characteristics of the signals obtained through Equation (1), i.e. are the signals more or less correlated after this operation? The authors also mention that, following this transformation of the SA module, "Residual connection is employed by integrating the input and output of the SA module", but it is not clear how much does this integrating procedure affect the spatial correlations between the EEG channels.

The graph attention mechanism proposed by the authors (page 5, Section 3.3.2) is similar to that of (Li et al., 2023).It would be interesting if the authors could explain the differences.
Li et al., "STGATE: Spatial-temporal graph attention network with a transformer encoder for EEG-based emotion recognition", Frontiers in Human Neuroscience, 2023

Regarding the intuition behind the proposed training methodology, the authors state that "The self-supervised strategy allows us to learn inherent patterns from EEG signals without labels, rather than directly separate different classes with supervised learning." (page 4, Section 3.2.3). This argument is not well-supported, as image embeddings extracted from pretrained models like ResNet, CLIP and ViT may in fact contain class-related information (Van Gansbeke et al., 2020).
"SCAN: Learning to classify images without labels", Van Gansbeke et al., ECCV 2020

Other comments:

There are some typos, grammar and syntax issues throughout the manuscript, e.g.:
- "We device" (page 2)
- "has remained a challenging." (page 2)
- "as the widely use way" (page 3
- "representation similarity anaylasis" (page 8)
- "categoring" (page 8)
- In Table 1, while s_1 is not presented in any layer's description, it is then used in the descriptions of output dimensions. I think that this is a mistake and that s_2 should be used instead.

This particular phrase is a bit vague and informal, the process could be stated more clearly: "we look for a few images that belong to the image concepts" (page 3)

**Questions:**

Important information about the experiments is missing from the manuscript. The number of trainable parameters for each EEG encoding architecture is not reported. Moreover, the dimensionality of the EEG features extracted from each EEG encoding architecture, exactly before being mapped to the dimensionality of the image features, should also be reported. This would allow the reader to obtain a better understanding on the details behind each architecture's performance. For example, it could be the case (among other reasons) that some EEG encoding architectures have better performance because of a larger EEG feature dimensionality (hence an EEG-to-image feature mapping layer of larger learning capacity).

The results presented in Figure 3 are not sufficiently explained. It is unclear to what do rows and columns correspond, i.e. which dimension corresponds to EEG features and which corresponds to image features. It would also be helpful to get an idea of the similarities between the visual features from (some of the) concepts of the training set and the visual features from concepts of the test set.

In Figure 2, sub-figure (D) is not sufficiently explained, i.e. the exact method that was used for obtaining the time-frequency map is not mentioned, the quantity that is depicted is not stated and it is unclear to what does the range [-1, 0] correspond.

---

> ### Author Response · Authors · 2023-11-19
> **Response to Review 9bdz**
>
> Thanks a lot for your expertise and thoughtful comments!
>
> We have to clarify here that the architecture of the EEG encoder was totally not the main contribution of this work, as mentioned at the end of Sec. 1. Our contributions lie in proving the feasibility of EEG-based object recognition, due to EEG provides low-cost, portable attributes with high time resolution, compared to fMRI and MEG. For this reason, we propose the self-supervised framework with nice classification performance and give evidence from different aspects, including detailly resolving temporal, spatial, spectral, and semantic aspects.
>
> Weakness
>
> W1. The architecture of EEG encoder.
>
> A1. We mentioned that we adapted the EEG encoder we used as the widely use way in EEG analysis in the third paragraph of Sec. 3.1 and named it the basic temporal-spatial convolution (TSConv). TSConv was simply adapted based on the data. The comparison in Table 3 showed the effectiveness of the framework with different EEG feature extractors.
>
> The encoder we used and the original ShallowNet encompassed several differences, mainly including the kernel size, activation function, and conv-pooling order. We compared these factors based on the final TSConv in the below table (**Appendix A.6**, Table 9). The three cases caused a significant effect on the top-1 accuracy separately compared to TSConv (p < 0.01, Bonferroni correction). We don’t want to prove which is better, but believe that any other well-designed EEG feature extractors can help us improve the performance in this framework.
>
> | Condition | top-1 acc (std) | top-5 acc (std) |
> | --- | --- | --- |
> | original ShallowNet | 7.9 (4.3) | 24.2 (9.3) |
> | kernel size as ShallowNet | 9.9 (2.4) | 33.5 (5.4) |
> | pooling order | 10.9 (3.0) | 34.3 (7.2) |
> | ELU->ReLU | 11.0 (2.5) | 33.3 (6.6) |
> | TSConv | 13.8 (3.3) | 39.5 (6.5) |
>
> W2. The spatial features captured by spatial modules, self-attention (SA), and graph attention (GA).
>
> A2. We applied the two spatial modules for two reasons. (1) enhancing the performance by maintaining spatial features. (2) using the trained model to give implied evidence for the brain processing. In Table 2, we have proved that SA and GA improved the top-1 by 0.9% (p < 0.05) and 1.8% (p < 0.01), separately. Besides, we used grad-CAM in Fig. 4(C) and Appendix A.3 Fig. 5(A) to show evidence that SA and GA help the model tend to temporal and occipital regions, which are related to visual object recognition processing. We also supply the attention weight visualization here to show a similar tendency in **Appendix A.3** Fig. 5(B).
>
> The residual connection helps to stabilize the training procedure habitually, especially for GA. We are sorry for neglecting to mention the use of residual connection for GA. In the table below (**Appendix A.6**, Table 8), we compare the effect of the residual connection. We can see that it significantly helps GA in top-1 accuracy (p<0.01) but has no significance for SA (p > 0.05).
>
> | Module | top-1 acc (std) | top-5 acc (std) |
> | --- | --- | --- |
> | SA w/ | res 14.7 (2.3) | 41.7 (4.6) |
> | GA w/ | res 15.6 (2.9) | 42.8 (5.5) |
> | SA w/o | res 13.8 (2.8) | 39.6 (5.7) |
> | GA w/o | res 10.1 (3.2) | 29.9 (5.3) |
>
> W3. The design and use of GA.
>
> A3. The STGATE is an interesting method that also uses graph attention networks for emotion recognition. Actually, both GA [R1] and SA [R8] in our application are very popular and usual ways to measure channel correlation, which introduces nothing new at the algorithm level. We would like to use these reliable ways to interpret the framework for learning plausible features.
>
> W4. Are labels used in this framework?
>
> A4. We only need the stimulus-response pairs of EEG and images to train the framework instead of the image labels. The ImageNet-1k and ImageNet-21k used to pre-train the ResNet and ViT were not the same as the 1654 image concepts in our training data. CLIP didn’t use labeled data for pre-training but image-text pairs [R9]. We can hardly perform a recognition task if the image features don’t have any class-related information.
>
> W5. Typos.
>
> A5. Thanks for the careful check! We have updated the drafts to deal with these typos.

---

> > ### Author Response · Authors · 2023-11-19
> > **Response to Review 9bdz**
> >
> > Question
> >
> > Q1. The design of different EEG encoders for comparison.
> >
> > A1. We have compared EEG encoders on the number of parameters and the output dimensions, as shown in the table below (**Appendix A.6**, Table 10). It’s not clear that the results are related to these two factors. We will further explore the EEG encoder in future works.
> >
> > | EEG encoder | Params. | Out dim |
> > | --- | --- | --- |
> > | ShallowNet | 102.0 k | 440 |
> > | DeepNet | 303.3 k | 1400 |
> > | Conformer | 161.2 k | 1440 |
> > | EEGNet | 12.8 k| 1248 |
> > | TSConv | 102.0 k | 1440 |
> >
> > Q2. Details of the semantic similarity analysis of Fig. 3.
> >
> > A2. We used representational similarity analysis (RSA) in Fig. 3(A) to show the category-level semantic information we obtained from EEG signals. The row denotes EEG features, and the column denotes image features.
> >
> > We plot t-SNE [R11] in **Appendix 7**, Fig. 7(A) to show the distribution between the visual features of the training and test set. We still select four categories (larger than concepts): animal, food, vehicle, and tool, with the top-down categorization provided in [R11]. The visualization reveals why we could achieve zero-shot performance after training with rich data. We also show the distribution between the EEG features of the training and test set in Appendix 7, Fig. 7(B).
> >
> > Q3. Details of the Fig. 2(D).
> >
> > A3. In Fig. 2(D), we average the channels belonging to different brain regions and use a short-time Fourier transform with Scipy to plot the time-frequency map. The x-axis denotes 1000 ms after the stimuli onset, the y-axis denotes the frequency, and the color bar unit is dB calculated by log. We would like to give a glance at the frequency distribution of the EEG signals, and then we perform quantitative analysis in Fig. 2(E).
> >
> > Thanks for your patience. Look forward to further discussion!
> >
> > [R1] Veličković, P., et al., Graph attention networks, ICLR, 2018. \
> > [R8] Vaswani, A., et al., Attention is all you need, NeurIPS, 2017. \
> > [R9] Radford, A., et al., Learning transferable visual models from natural language supervision, ICML, 2021. \
> > [R10] Van der Maaten, L., & Hinton, G, Visualizing data using t-SNE, JMLR, 2008. \
> > [R11] Hebart, M. N., et al., THINGS: A database of 1,854 object concepts and more than 26,000 naturalistic object images. Plos One, 2019.

---

> > ### Comment · Reviewer_ixpa · 2023-11-20
> > **shallownet nonlinearity confusion**
> >
> > What do you mean with ELU -> ReLU? The original shallownet (https://github.com/braindecode/braindecode/blob/22da938bec9445aaa7a7a58a6abcfb14bb2d1d71/braindecode/models/shallow_fbcsp.py) has squaring->avgpool->log as the sequence of operations,
> > see
> > https://github.com/braindecode/braindecode/blob/22da938bec9445aaa7a7a58a6abcfb14bb2d1d71/braindecode/models/shallow_fbcsp.py#L78-L80
> > https://github.com/braindecode/braindecode/blob/22da938bec9445aaa7a7a58a6abcfb14bb2d1d71/braindecode/models/shallow_fbcsp.py#L170-L178

---

> > > ### Author Response · Authors · 2023-11-20
> > > **ShallowNet Implementation**
> > >
> > > Thanks for your reference.
> > >
> > > In response to W1, we show that the activation function has a significant impact on the performance using a common alternative, ReLU, which was also mentioned in [15].
> > >
> > > Here, we provide the results using squaring with the ShallowNet and the adapted TSConv, as below. The results of both models largely decrease. There were many comparisons in [R15], where different settings may vary on different data. We think the ShallowNet has given us a very good reference to develop better encoders.
> > >
> > > | Condition | top-1 acc (std) | top-5 acc (std) |
> > > | --- | --- | --- |
> > > | ShallowNet w/ square | 5.1 (1.8) | 18.7 (5.2) |
> > > | TSConv w/ square | 6.8 (3.1) | 23.4 (6.2) |
> > >
> > > [R15] Schirrmeister, R. T., et al., Deep learning with convolutional neural networks for EEG decoding and visualization, HBM, 2017.

---

> > > > ### Comment · Reviewer_9bdz · 2023-11-22
> > > >
> > > > I would like to thank the authors for productively engaging in the rebuttal process and carefully updating the manuscript.
> > > > The answers to the questions regarding the attention mechanisms, the network architectures and the figures have been very helpful. The extra experiments and the preliminary analysis on MEG data also add value to this work. I do not have any further questions from my initial review that remain unaddressed. I will take all these into consideration for determining the final rating of the manuscript.

---

> > > > > ### Author Response · Authors · 2023-11-23
> > > > > **Response to Review 9bdz**
> > > > >
> > > > > Thanks a lot for your insightful reviews and positive feedback to help us improve this work!

---

### Official Review · Reviewer_JYif · 2023-10-30

**Soundness:** 3 good
**Presentation:** 3 good
**Contribution:** 3 good
**Rating:** 8
**Confidence:** 4

**Summary:**

The authors present a deep learning pipeline to retrieve the images seen by participants whose brain activity was collected using EEG. The pipeline combines a pretrained frozen image encoder (CLIP, ViT or ResNet) and an EEG encoder (a parameter efficient ConvNet optionally prepended with a spatial attention layer) trained with a contrastive loss and evaluated in a retrieval scenario. Results favorably compare to an existing baseline and to existing neural network architectures. An analysis of the decoding performance and of the biological plausibility of the results highlights spatial, temporal and spectral patterns underlying the image decoding task.

**Strengths:**

Originality: The proposed approach combining a frozen image encoder and a trained brain encoder has previously been used on fMRI, however it appears its application to EEG data is novel.

Quality: The manuscript is overall of good quality. The experiments adequately answer the core research question of the paper and provide supporting evidence for the physiological plausibility of the decoded patterns (i.e. through the complementary analyses of the decoding performance, spatial/temporal/spectral patterns and impact of number of stimulus repetitions).

Clarity: The manuscript is clearly written, and most figures and results are presented clearly.

Significance: This study goes beyond the existing image decoding results from EEG in which a limited number of classes were considered. It also is an important building block toward better image decoding from EEG, following in the footsteps of previous studies in fMRI.

**Weaknesses:**

- Some of the secondary claims do not appear to be adequately supported by the presented results. For instance, the claims on gamma-based decoding (see Q3 below) and on the use of occipital information in the graph attention layer (Q4).

- Based on previous image decoding work in fMRI and EEG, a logical continuation of the proposed contrastive learning pipeline is to then pass predicted latents to a generative image model (see Q5). I believe the presented image decoding performance results (along with the analyses of Sections 4.5-4.8) are interesting in themselves, however as a result the scope of the paper is narrower than existing studies on the topic. I believe the authors should discuss this in their manuscript.

**Questions:**

Q1. In Section 3.1, the authors write “These images are processed and averaged to obtain one template for each concept.” It is not clear to me how these averaged images are used in the experiments - was this in the semantic similarity analysis? Moreover, were the images averaged in pixel space or in latent space?

Q2. In Figure 2A: What is plotted on the topomaps? I assume it is the average EEG amplitude of each channel across the 100 ms window. If so, how was the data normalized (since the colorbar is from -1 to 1)? Also, in Figure 2B: What do the shaded areas represent? Is it variability across subjects?

Q3. In Section 4.5: “High-frequency responses could be observed from electrodes on the temporal cortex.” How is that measured? I find it hard to conclude from visual analysis of Figure 2D. Along with the chance-level performance of the gamma band (Figure 2E) I don’t see compelling evidence for the following claim: “These results align with established principles, coarsely indicating that the bottom-up feedforward is carried by the synchronization of theta and gamma bands.”

Q4. In Figure 4C and in Suppl. Figure 1: If I understand the topomaps correctly it looks like graph attention actually mostly ignored occipital electrodes for most subjects (as opposed to self-attention for which the occipitally-located pattern is very clear). It would be interesting to look at the average attention weights predicted by both attention modules instead of using GradCAM for this kind of analysis. More generally, given the two formulations (self-attention and graph attention) end up predicting C weights, directly comparing their respective attention weights might give more insight into what drives their different performance. Finally, a central difference between the two approaches appears to be the use of a residual connection after the attention module - it would be interesting to evaluate whether that explains (part of) their differences.

Q5. Existing image decoding studies, e.g. in fMRI (Ozcelik & VanRullen, 2023) but also in EEG (Palazzo et al., 2020), have extended a similar approach up to image generation by feeding predicted image latents to a generative image model. Have the authors attempted this step?

Ozcelik, Furkan, and Rufin VanRullen. "Natural scene reconstruction from fMRI signals using generative latent diffusion." Scientific Reports 13.1 (2023): 15666.
Palazzo, Simone, et al. "Decoding brain representations by multimodal learning of neural activity and visual features." IEEE Transactions on Pattern Analysis and Machine Intelligence 43.11 (2020): 3833-3849.

Q6. The description of the analysis of Section A2 is a bit confusing. The comparison appears to be made between (1) using pretrained frozen image encoders and (2) using randomly initialized image encoders which are trained along the EEG encoders. The description doesn’t make it clear that the difference in computational resources between both approaches comes from the frozen/unfrozen dimension, not from the pretrained/non-pretrained dimension.

Q7. Algorithm 1 describes the CLIP loss (Radford et al., 2021). I think this should be introduced and cited as such, rather than through the lens of InfoNCE.

Radford, Alec, et al. "Learning transferable visual models from natural language supervision." International conference on machine learning. PMLR, 2021.

---

> ### Author Response · Authors · 2023-11-19
> **Response to Review JYif**
>
> Many thanks for your detailed and insightful suggestions! We are glad to see you approved the contributions of our work.
>
> Q1. The use of templates with averaged images.
>
> A1. In the test phase, the incoming EEG trial was processed by the EEG encoder and then compared the cosine similarity with image templates. Therefore, we used several images for each image concept to construct the templates by averaging the image features at the latent level.
>
> It’s worth noting that we preferred to define it as a classification task, because we didn’t use stimulus images for the templates but other images from the corresponding concepts.
>
> Q2. Explain the topomaps in Fig. 2(A) and shade in Fig. 2(B).
>
> A2. Fig. 2(A) plotted EEG amplitude with a 100 ms window. We performed z-score standardization on the raw data and then normalized to [-1, 1]. In Fig. 2(B), the shade denoted the standard deviation of different subjects.
>
> Q3. Analysis of Gamma components.
>
> A3. It could be seen in Fig. 2(D) that the high-frequency responses in the temporal area were slightly more obvious than in the occipital and parietal areas. But the performance indeed achieved a chance level, where gamma was known for being related to object recognition. Two possible reasons were attached at the end of Sec. 4.5. (1) There was a longstanding controversy about whether scalp EEG can obtain high-frequency response due to artifacts like EMG [R4] [R5]. (2) Gamma had a small amplitude and easily to be modulated by other cognitive processing. Besides, the current dataset uses rapid sequences for stimuli, which makes the responses interfered by the pre- and post-stimulus. We plan to evaluate longer stimulus duration in future work to explore gamma responses. We have also mentioned the limitation in Sec. 5.
>
> Q4. Attention weights visualization of self-attention (SA) and graph attention (GA).
>
> A4. We applied two popular ways to design “spatial filters” in the EEG encoder for two reasons. (1) enhancing the performance by maintaining spatial features. (2) using the trained models to give implied evidence for the brain processing. As in Fig. 4(C) and the supplements, we could see that the spatial modules tended to the occipital area (GA) and parts of the temporal area (SA).
>
> We have visualized the attention weights of SA and GA with all training data of each subject, as shown in **Appendix A.3**, Fig. 5. Interestingly, more clear trends in occipital and temporal areas can be observed. However, we have to admit that it’s an intuitive way to use trained models to reflect the brain signal response. We still need other rigorous evaluations.
>
> Actually, both SA and GA used a residual connection, which was a mistake in the draft. We have made a comparison of the residual connection in the Table shown below (**Appendix A.6**, Table 8). We can see that residual connect significantly helps GA in top-1 accuracy (p<0.01) but not for SA (p > 0.05). The top-1 accuracy of GA is not significantly better than SA when both of them use the residual connection (p > 0.05). We will provide the source code later for reproduction.
>
> | Module | top-1 acc (std) | top-5 acc (std) |
> | --- | --- | --- |
> | SA w/ res | 14.7 (2.3) | 41.7 (4.6) |
> | GA w/ res | 15.6 (2.9) | 42.8 (5.5) |
> | SA w/o res | 13.8 (2.8) | 39.6 (5.7) |
> | GA w/o res | 10.1 (3.2) | 29.9 (5.3) |
>
> There are other differences between these two methods, such as the way for attention calculation, the activation function, etc. We will attempt to go deeper into designing attention or graphs to help us perceive the signal dynamic in future works.
>
> Q5. Why not image generation?
>
> A5. We know that there are some works performing image reconstruction, even caption with brain signals, especially fMRI [R5] [R6]. Brain features are extracted and fed into well-designed generators like diffusion models.
>
> Although exquisite images are decoded, we are not easy to figure out what we really obtained from brain signals (low-level or high-level information?), even though some works separate structural and semantic features [R7] and prove them with different metrics.
>
> On the other hand, we think one of the most important gains is the concept/category-level semantic discrimination when visual decoding goes from low-level stimuli (contrast, color, orientation, etc.) to high-level stimuli (natural images). Therefore, we would like to begin with the recognition task to prove the feasibility of EEG signals. We have also mentioned it at the beginning of Sec. 5.
>
> Q6. The description for evaluating the pre-trained effect in Sec. A2.
>
> A6. The increment of the computational cost was from training image encoders. Using pre-trained image encoders, we can process the images and obtain features before training the framework. Thus, the computational cost and training time of using pre-trained image encoders would be much lower than randomly initialed ones, which is meaningful for real-world use. We have updated the description for clarity.

---

> > ### Author Response · Authors · 2023-11-19
> > **Response to Review JYif**
> >
> > A7. Clear introduction of contrastive loss.
> >
> > Q7. We have updated the drafts for clear reference.
> >
> > We enjoy these pertinent suggestions and look forward to further discussion!
> >
> > [R4] P. Fries, et al., Finding gamma, Neuron, 2008. \
> > [R5] S. Yuval-Greenberg, et al., Transient induced gamma-band response in EEG as a manifestation of miniature saccades, Neuron, 2008. \
> > [R5] Takagi, Y., et al., High-resolution image reconstruction with latent diffusion models from human brain activity, CVPR, 2023. \
> > [R6] Ozcelik, F., et al., Natural scene reconstruction from fMRI signals using generative latent diffusion. Scientific Reports, 2023. \
> > [R7] Fang, T., et al., Reconstructing perceptive images from brain activity by shape-semantic gan, NeurIPS, 2020.

---

> > > ### Comment · Reviewer_JYif · 2023-11-21
> > >
> > > Thanks to the authors for their answers and clarifications.
> > >
> > > [Q3] I still find the qualitative trend of Figure 2D, along with the chance-level performance of the gamma band in Figure 2E, very limited evidence for a wide claim on bottom-up and top-down processing. Can the authors clarify how their observations on the theta, alpha, beta and gamma bands align with “established principles”?
> > >
> > > [Q4] Thank you for providing figures with the attention weights. Interesting to see that the patterns look quite different from those found with GradCAM!
> > >
> > > [Q6] This is nitpicking, but my point was that the reason pre-trained image encoders are more computationally efficient is that they are *frozen*, not that they are pre-trained. If you were to finetune them (instead of keeping them frozen) along with the EEG encoder, you would end up with the same number of learnable parameters and same memory usage as randomly initialized image encoders in Table 5. Therefore, adding a column in Table 5 to specify whether each model is frozen (i.e. the features are extracted beforehand) or finetuned might clarify this point.

---

> ### Author Response · Authors · 2023-11-21
> **Response to Review JYif**
>
> Thanks for your feedback!
>
> [Q3] It’s indeed an overambitious statement with the time-frequency map, which only shows a trend. We have deleted this sentence from the draft.
>
> [Q4] Yes, we also think it’s interesting to observe brain responses with deep learning models, though the current version is a qualitative attempt.
>
> [Q6] You are right. The decrease in computational cost comes from the freezing of the image encoder. We have updated the content and footnote of Table 5 and the description to clarify that.
> > In A.2: We should mention that the decrease in computational cost actually comes from the freezing of the pre-trained image encoder and getting image features beforehand.
>
> > Table 5 footnote: pre-trained ResNet-50 and ViT-B/16. The image features were obtained before training.
>
> We are very grateful for your patience in helping us improve the work!

---

> > ### Comment · Reviewer_9bdz · 2023-11-22
> > **Clarification about Fig.2**
> >
> > In their response about Q2, the authors state the following: "Fig. 2(A) plotted EEG amplitude with a 100 ms window. We performed z-score standardization on the raw data and then normalized to [-1, 1]."
> >
> > If the case is that z-scoring and normalization are applied within each 100msec window, then the authors should further explain this choice. Otherwise, I would suggest  the authors to consider the option of not applying these transformations per 100msec sub-window -- doing that on the entire plotted length instead.

---

> ### Author Response · Authors · 2023-11-23
> **Response to Confusion on Fig. 2**
>
> Thanks for your careful check!
>
> We preprocessed the data first and then plotted the topographies in the visualization of EEG and MEG data. The colorbar on the right side is applied to the ten subfigures. We can see a gradual trend in the responses to the stimulus image.
>
> We would like to mention that this dataset leveraged rapid event sequences consisting of 20 images with a stimulus onset asynchrony (SOA) of 200 ms (100 ms stimulus with 100 ms blank screen). The rapid serial visual presentation (RSVP) helped to obtain rich and large datasets. However, it also took more complex brain responses, such as periodic changes in the occipital region and gradual increase in the frontal region. Using a longer SOA (500 ms stimulus with 1000±200 ms jitter blank screen) like the MEG data could be a better way to observe the visual response to one image.

---

### Official Review · Reviewer_ixpa · 2023-11-05

**Soundness:** 4 excellent
**Presentation:** 3 good
**Contribution:** 3 good
**Rating:** 8
**Confidence:** 3

**Summary:**

The manuscript aims to match images to the EEG signals they evoke using a contrastive loss on the encodings of an EEG model and a vision model. The models are evaluated on zero-shot-discriminating 200 images, that were shown 80 times each, and the corresponding 80 EEG signals are averaged for each image.  The manuscript introduces an EEG model specific for this task and results show it to reach improved decoding performance over existing commonly used EEG models. Further, the manuscript finds that semantically similar images of the same category (like animal, food) have more similar learned EEG representations compared to images of different categories. Temporal analysis shows 100 to 600ms to yield the highest decoding accuracy. Frequency analysis show delta theta and beta bands contain information for reasonable decoding accuracies. The manuscript also performs a variety of spatial analyses.

**Strengths:**

* The manuscript is well-written
* method is straightforward to understand
* a lot of detailed analysis of the results on different levels
* helpful figures

**Weaknesses:**

Some details I found not easy to understand, see below

**Questions:**

I am not sure I understood the graph attention method details, I found the notation a bit confusing, does aT mean the result fo applying the feedforward layer is transposed? Also it would be nice to mention if this is a unique way of doing graph attention or a well-established way, then would be good to provide references for it.

For the GradCam Analysis, I am not sure I understood exactly which gradients of which layer do you use and with regard to what? These details may be helpful to write more clearly.

A very recent work tackling similar problems is https://arxiv.org/abs/2310.19812. Due to its recency, I do not expect any comparison, but a brief mention of this work as concurrent work would be great.

Font in Figure3 b could be bigger
Figure 4 Colored stars could be bigger


Typo p. 8: anaylasis

---

> ### Author Response · Authors · 2023-11-19
> **Response to Review ixpa**
>
> Thank you for your positive evaluation of this work and thoughtful comments!
>
> Q1. The use of graph attention.
>
> A1. In equation (2), $a$ denotes a single-layer feedforward layer for attention calculation, which is parametrized by a weight vector $a \in \mathbb{R}^{2T}$, where $T$ represents the time length of EEG trials. The transposition operator is used for calculating with the concatenated node vectors.
> The graph attention here is from popular graph attention networks (GATs) [R1] and implemented following [R2]. We only gave a reference at the end of the fourth paragraph of the introduction. We have updated the draft for clarity.
>
> Q2. Gradients of Grad-CAM.
>
> A2. In the paper, we applied self-attention (SA) and graph attention (GA) to capture spatial features in the EEG signals for (1) enhancing the EEG encoder with easily used tricks, and (2) digging the features for evidence of capturing spatial dynamics of human object recognition. Therefore, the gradients were from the whole SA or GA layer.
>
> Q3. MEG work comparison.
>
> A3. We also followed this interesting paper using MEG signals for image decoding recently. It proves the feasibility of this task with MEG, which relies on time resolution. It’s a good way to confirm our work with EEG data further.
>
> Here, we struggle to supply some preliminary results in **Appendix A.5** with the MEG data used in [R3], including (1) the results of 200-way zero-shot classification and retrieval tasks (A. 5, Table 7), and (2) temporal, spatial, and spectral analysis (A.5, Figure 6) to show consistency with EEG results. We find that our retrieval performance is lower than that in [R3], possibly due to (1) they leverage the data from other subjects to train the model of a specific subject, and (2) Our model is designed for EEG signals and is not optimized for MEG, whose channel dimensions are much larger. However, we believe our analysis will give a reference to future works.
>
> Q4. Figures and typos.
>
> A4. We have fixed these issues in the draft.
>
> Look forward to any feedback following this response!
>
> [R1] Veličković, P., et al., Graph attention networks, ICLR, 2018.\
> [R2] Brody, S., et al., How attentive are graph attention networks? ICLR, 2022.\
> [R3] Benchetrit, Y., et al., Brain decoding: toward real-time reconstruction of visual perception, arXiv: 2310.19812.

---

> > ### Comment · Reviewer_ixpa · 2023-11-20
> >
> > Thanks for your answers and revisions.
> >
> > I still think would be good to be even more explicit about the fact that you are using an existing graph attention module, like "The process to update one electrode is as follows, using the GA from Velickovic et al. (2018)"
> >
> > Also, I think there is an inconsistency between your reply here and the formula in the manuscript, you mention you implemented following [R2], but the formula 3 seems to be from R1? see https://arxiv.org/pdf/2105.14491.pdf p.5 formulas 6 and 7

---

> > > ### Author Response · Authors · 2023-11-20
> > > **Response to the GAT Description**
> > >
> > > Thanks for your suggestion! Sorry for the confusion. We got the idea from [R1] and referred to the code implementation in [R2], which has the same definition of GAT but a different attention operation order.
> > >
> > > We have added the description in Sec 3.3.2, as “We employ the GA module to update each electrode with all the others, using the implementation from Brody et al. (2022),” and also adjusted the formula.

---

### Author Response · Authors · 2023-11-22
**General Response**

Many thanks to all our four reviews for the helpful and thoughtful comments!

In this work, we have demonstrated the feasibility and biological plausibility of EEG-based image decoding from three folds: (1) zero-shot classification performance achieved by a self-supervised framework, (2) detailed resolving of the brain activity from temporal, spatial, spectral, and semantic aspects, and (3) model interpreting revealed by two spatial modules.

We are pleased that reviews have confirmed our originality and good quality, such as well-written drafts, a lot of detailed and interesting analysis, adequate experiments to answer the core research question, etc. We have made substantial complements in response to reviewers’ concerns, including (1) attention weights visualization of spatial modules, (2) preliminary analysis on a MEG dataset for reference, (3) implementation comparison of EEG encoders, (4) features visualization, and (5) detailed explanation of confusions.

Please see the updated draft [here](https://openreview.net/pdf?id=dhLIno8FmH) and the highlighted version in the [supplementary material](https://openreview.net/attachment?id=dhLIno8FmH&name=supplementary_material).

Thanks again for your time and patience in helping with this work! We are looking forward to any further feedback.

---

### Meta-Review · Area_Chair_9RKU · 2023-12-14

**Metareview:**

This paper presents an EEG-to-image decoder based on contrastive learning on paired EEG and image encoders. The model is trained and demonstrated on real-world datasets. The authors further develop the scientific potential of their approach by analyzing the EEG features learned by their model as related to the image dataset. The reviewers all appreciated the approach, with mostly concerns on clarity. There were some concerns on novelty and relationship to the literature raised by one reviewer, which while I do not think were completely addressed by the authors is a minor concern given the remainder of the positive comments and potential contribution. Therefore I recommend that this work be accepted.

**Justification For Why Not Higher Score:**

There are some concerns that were not fully addressed by Reviewer 4 related to the framing of the past literature and some of the interpretations of the scientific results. I do believe these points have merit, however do not overturn the overall positive assessment and potential recognized by the other reviewers.

**Justification For Why Not Lower Score:**

Despite the concerns raised by Reviewer 4, the model does show potential in both decoding EEG and in understanding the signal content.

---

### Decision · Program_Chairs · 2024-01-16

Accept (poster)